# Large eddy simulation of boundary-layer turbulence over the heterogeneous surface in the Source Region of the Yellow River

Yunshuai ZHANG[a], Qian HUANG[a,*], Yaoming MA[a,b,c,d], Jiali LUO[a], Chan WANG[e], Zhaoguo LI[e], and Yan CHOU[a]

[a] *Key Laboratory for Semi-Arid Climate Change of the Ministry of Education, College of Atmospheric Sciences, Lanzhou University, Lanzhou 730000, China*
[b] *Key Laboratory of Tibetan Environment Changes and Land Surface Processes, Institute of Tibetan Plateau Research, Chinese Academy of Sciences, Beijing 100101, China*
[c] *University of Chinese Academy of Sciences, Beijing 100049, China*
[d] *CAS Center for Excellence in Tibetan Plateau Earth Sciences, Chinese Academy of Sciences, Beijing 100101, China*
[e] *Key Laboratory of Land Surface Process and Climate Change in Cold and Arid Regions, Northwest Institute of Eco-Environment and Resources, Chinese Academy of Sciences, Lanzhou 730000, China*

## Abstract

Lake breezes are proved by downdrafts and the divergence flows of zonal wind in the source region of the Yellow River (SRYR) in the daytime based on ERA-Interim reanalysis data. In order to depict the effect of the circulations induced by surface anomaly heating (patches) on the boundary-layer turbulence, the large eddy model was used to produce a set of 1D strip-like surface heat flux distributions based on observations, which obtained by a field campaign in the Ngoring Lake Basin in the summer of 2012. The simulations show that for the cases without ambient winds, patch-induced circulations (SCs) enhance the turbulent kinetic energy (TKE) and then modify the spatial distribution of TKE. Based on phase-averaged analysis, which

*Corresponding author.
E-mail address: huangq@lzu.edu.cn (Q. Huang).



separates the attribution from the SCs and the background turbulence, the SCs
contribute no more than 10% to the vertical turbulent intensity, but their contributions
to the heat flux can be up to 80%. The lake patches produce consistent spatial
distributions of wind speed and turbulent stress over the lake–land boundary, and the
obvious change of turbulent momentum flux over the boundary of patches can not be
neglected. In the entrainment layer, the convective rolls still persist under stronger
geostrophic winds of 7–11 m s$^{-1}$. The increased downdrafts, which mainly occur over
the lake patches and carry more warm, dry air down from the free atmosphere. In
general, the SCs promote the growth of convective boundary layer, while the
background flows inhibit it. The background winds also weaken the patch-induced
turbulent intensity, heat flux, and convective intensity.
**Key word:** turbulence, heat flux, heterogeneously surface heating, background flows,

phase-averaged analysis

## 39   1. Introduction

Turbulence in the planetary boundary layer (PBL), which is derived from surface
heating and surface fraction, plays an important role in the exchange of heat,
momentum, moisture, and chemical constituents between the surface and free
atmosphere (Zhang et al., 2018). Previous studies on the turbulence and turbulent
exchange over homogeneous surfaces based on Monin-Obukhov similarity theory
were conducted before the 1990s (Sommeria and LeMone, 1978; Moeng, 1984).
Turbulence over heterogeneous surfaces was investigated through field campaigns
(Wang et al., 2016; Zhao et al., 2018) and numerical simulations (Shao et al., 2013;



Liu et al., 2011) in the past few decades, which has improved our understanding of the
transfer and spatial and temporal variability of the turbulence. Thermal surface
heterogeneity is a typical issue and leads to the formation of local/secondary
circulations. Sea and lake breezes are a well-known example of flows that are
generated by heterogeneous surface heating between the land and water (Crosman and
Horel, 2012). Observations have also revealed the imbalance in the surface energy
budget over heterogeneous surfaces (Foken et al. 2010; Xu et al., 2016). The most
widely used eddy covariance (EC) system for a single site has been shown to
underestimate the turbulent flux due to the large-eddy transport or secondary
circulations not being captured (Foken et al., 2010; Xu et al., 2016). The simulation
studies conducted by Zhou et al. (2018) and Frederik and Matthias (2018) showed that
the flux induced by mesoscale or secondary circulations is the main reason for the
energy imbalance. Moreover, the PBL parameterization schemes in the
weather/climate model over a heterogeneous surface have been continuously
improved until now (Avissar and Pielke, 1989; Shao et al., 2013). Different surface
patterns such as mosaic (Avissar and Schmidt, 1998), chessboard (Liu et al., 2011;
Shen et al., 2016), patchy-like (Zhou et al., 2018), and strip-like (Li et al., 2011;
Wang et al., 2011) patterns have been utilized to simulate thermodynamic surface
heterogeneity. These studies confirmed that the secondary circulation induced by the
surface heterogeneity influences the PBL's properties and turbulent characteristics. In
addition, several studies have examined the effects of surface heterogeneity on
different levels of background winds (Shen and Leclerc, 1995) and the direction



relative to the orientation of the heterogeneity (Wang et al., 2011; Kang and
Lenschow, 2014). However, the issues related to the effects of the surface
heterogeneity in special areas still need to be explored.
The dynamic and thermodynamic influences of the Tibetan Plateau (TP) on the
regional and global weather and climate systems are closely related to its PBL, and
turbulence plays a significant role in the mass and energy exchange between the TP
and the atmosphere (Chen et al., 2013; Chen et al., 2016). Different landscapes make
up the heterogeneous land surface over the TP. As the Asian Water Tower, lakes are
widely and densely distributed over the TP, which affects the overlying energy and
mass transport through the lake-air turbulent heat flux. Biermann et al. (2014) and
Wang et al. (2015) discovered that the turbulent flux of Lake Nam co, which is
surrounded by wet grasslands, is actually very considerable but was often
underestimated in the model. The Source Region of the Yellow River (SRYR) is
located in the northeastern part of the TP and is known as the "water tower" of China
because it contains 48 lakes. The Ngoring and Gyaring (Sisters) Lakes are two major
lakes, and Ngoring Lake is the largest in the SRYR (Li et al., 2015; Wen et al., 2015).
In addition to the lakes, the forests, alpine meadow, wetlands, rivers, and glaciers
comprise the diverse underlying surfaces in the SRYR, with grassland accounting for
about 80% of the area (Mudassar et al., 2018). Consequently, the SRYR is an ideal
region for studying the turbulence over a heterogeneous land surface.
Observational studies have revealed that water vapor, heat, and energy exchange
occur over alpine meadows/wetlands (Zheng et al., 2015; Jia et al., 2017) and lakes





(Li et al., 2015; Wen et al., 2016), and models have been used to simulated the effects
of the lakes on the cool and moist regional climate (Wen et al., 2015; Ao et al., 2018).
However, the features of the boundary-layer turbulence over the heterogeneous
underlying surfaces and the effects of thermodynamic surface heterogeneity on the
turbulent flux in the SRYR remain unclear. Over the last few decades, the lakes have
shrunk and the grasslands have degraded in the SRYR due to climate change and the
excessive utilization of water resources (Brierley et al., 2016; Mudassar et al., 2018).
It is essential to investigate the variation in the structure of boundary-layer and
turbulent heat flux with changes in the surface's thermal properties and the
background winds.
High resolution field measurements are extremely rare on the TP because of the
harsh environmental conditions, so few observational studies on the turbulence
characteristics and the turbulent heat flux have been conducted. Large eddy
simulation (LES) has the unique advantage of being accurate and able to describe
turbulence finely, and thus, it has been widely used to investigate the effects of
surface heterogeneity on turbulence (Hadfield et al., 1991, 1992; Kang and Lenschow,
2014). However, little has been done to improve our understanding of how the surface
heterogeneity affects the boundary-layer turbulence, and the contributions of the
patch-induced motions to the turbulent flux and energy in the SRYR. Furthermore,
modeling the turbulence over the heterogeneous surface in the SRYR can not only lay
a basis for the analysis of the local energy and mass transport, but it can also provide a
quantitative    reference    for    improving    the    parameterization    schemes    over



heterogeneous surfaces in weather and climate models.
In this study, we used LES to investigate the detailed turbulence characteristics in
the SRYR. Our primary focus was the impacts of the surface heat flux anomalies on
the turbulent kinetic energy (TKE), turbulent intensity, and turbulent flux. The
turbulence characteristics and turbulent fluxes in the surface and the entrainment
layers were investigated, too. This paper is arranged as follows Section 2 describes
the model and data used in this study. Section 3 discusses the modeled results in detail,
and section 4 provides a summary and discussion of our findings.

## 2    data and methods

### 2.1   Study area and observations

Ngoring Lake and Gyaring Lake (hereinafter referred to as the two lakes) are
located in the SRYR and are surrounded by the alpine meadow. Their mean elevation
is 4274 m above sea level. The study area is shown in fig.1. The turbulent flux and
standard atmospheric variables were measured over the lake and grassland surface.
The GPS radiosonde data from the field campaign on July 29, 2012, 30 m west of
Lake Ngoring (near the gradient tower station, TS) and at Madoi station (MD) located
30 km the east of the lake (34.918° N, 98.216° E, 4279 m AMSL), as well as the eddy
covariance data for Lake Station (LS) above the northwest of the lake (35.026° N,
97.652° E) and Grassland Station (GS) (34.913° N, 97.553° E) 1.5 km west of the
lake shore were used. For further details on the field campaign and the quality control
of the sounding and eddy covariance data, see Li et al. (2015) and Li et al. (2017).
The synoptic background near the surface and at 500 hPa and the distribution of the
wind components in the vertical and horizontal directions were investigated using the
ERA-Interim Reanalysis Data with a 1° × 1° resolution collected at 12:30 LT and
18:30 LT (LT: local time, used in the whole study) on July 29, 2012, with a delimiting
a range of 32° N–37° N, 95° E–100° E, including the two lakes area (34.8° N–35° N,
97° E–98° E) and the surrounding grassland.

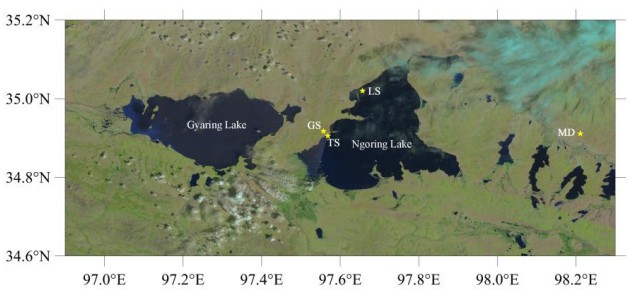

**Fig.1.** Map of the study area obtained using Landsat data, with the location of the observation
stations marked by yellow stars. The turbulent fluxes were measured at LS and GS stations. The
standard atmospheric variables were observed at the TS. MD is a fixed meteorological observatory
of the China Meteorological Administration.
**2.2   Methods**
**2.2.1   Simulations set-up over the heterogeneous underlying surface**
The U.K. Met Office large eddy model (LEM) version 2.4 (Gray et al., 2001) was
used in this paper. The LEM is a three-dimensional and non-hydrostatic numerical
model, which can be used to simulate a wide range of turbulence-scale and
cloud-scale problems with a high resolution. The domain size was 135 km × 30 km ×
6 km with a horizontal grid-spacing of 200 m. A vertically stretched grid with a
minimum spacing of 1.1 m was utilized in the surface layer and a maximum of 64.8 m
above 2000 m. Periodic lateral boundary conditions were applied, with a rigid lid at
the top of the model domain. To reduce the reflection of the internal gravity waves, a


Newtonian damping layer was applied above 3500 m. The surface boundary
conditions of the model were derived from the Monin-Obukhov similarity theory
using the Businger-Dyer functions. The subgrid model used in the LEM was based on
the Smagorinsky-Lilly approach (Brown et al., 1994). The potential temperature, wind
($u$ and $v$), and relative humidity profiles obtained during the field campaign on July 29,
2012, were used to initialize the 3D runs. The LEM was driven by the time-varying
sensible heat flux (SHF) and the latent heat flux (LHF) at the surface. The geostrophic
wind shear was calculated using the ERA-Interim geostrophic wind at the surface and
at 1500 m. The simulation time was 12 hours and the data were output every hour. In
this study, twelve 3D runs with different surface heat fluxes under various ambient
wind conditions were performed. Two of the runs were horizontally homogeneous
with a uniform grass surface under the conditions of wind (HOMW) and no wind
(HOM). The other cases were simulated with one (A1L) or two (A2L) lake patches in
the middle of the model domain. The surface heat flux anomaly was applied over a 30
km wide strip (two strips for A2L) extending the entire 30 km width of the domain in
the x-direction. Here the term heat flux refers to both the sensible and latent heat
fluxes. This can be viewed as representing Ngoring Lake and Gyaring Lake in the
SRYR. It should be noted that in this study the scale of the heterogeneity was large
enough to enable the formation of small eddies over the lake patches that could
coexist with the large-scale patch-induced circulations (Patton et al., 2005). Four
simulations (A1L, A2L, A1LW, and A2LW) were initialized using the surface heat
flux over the patch/patches measured at LS and the heat flux outside the patch/patches





measured at GS. This means that the heat flux into the modeled atmosphere decreases
as the number of patches increases. However, it is helpful to separate the effects of the
total increase in heating from the effects of the localization of the heating when
considering the consequences of an unresolved spatially changing heat flux for a
global model. In order to keep the total heat flux in the modeled atmosphere constant,
a "balanced" surface heat flux approach was used. Therefore, if the surface heat flux
observed at the GS is denoted as FGS, and the heat fluxes over the patch and outside
of the patch are denoted as FL and FG, respectively. FL and FG were calculated using
the following equations:
$$FL = FGS \times \left( SL/ST \right) \tag{1}$$

$$FG = FGS \times \left( SG/ST \right) \tag{2}$$

where ST, SL, and SG are the squares of the model domain, the patch, and the outside
of it, respectively. Another four simulations (A1L_C, A2L_C, A1LW_C, and
A2LW_C) were performed using this balanced surface heat flux approach. The
heterogeneous initial conditions were used in the surface heat flux anomaly
simulations. The initial profiles over the patch/patches were derived using the data
from TS station, and the data observed at the MD station used for the outside
patch/patches. Various ambient wind conditions were also used for the surface heat
flux anomaly runs. The parameters and the conditions of each run are listed in Table 1
for convenience. Sketches of the heterogeneous surface and of the surface heat fluxes
over the lake patches and the outside patches for the unbalanced and balanced cases
are depicted in fig. 2.



The initial potential temperature and special humidity are shown in fig. 2h, and the
horizontal components of the wind profiles and the geostrophic wind are shown in fig.
2g. A stable layer was found over the grass and a 200 m convective boundary layer
(CBL) was found over the lake at 06:30 LT. The special humidity profiles show that
the air tends to be moister over the lake (dash lines in fig. 2h). The study area is
characterized by a considerable surface heat flux and high wind speeds in the daytime.
The stronger initial velocity is from the GS, which recorded wind speeds of up to 10
m s$^{-1}$ below 500 m.
**Table 1**
Parameters for the 3D simulations.

| Name | Wind field | Surface heat flux (SHF and LHF) | Number of lake patches | Size of Lake patch (km) |
|---|---|---|---|---|
| HOM | without wind | FGS | 0 | - |
| HOMW | initial wind + geostrophic wind | FGS | 0 | - |
| A1L | without wind | lake patch: FLS (Heat flux that observed at LS); outside patch: FGS | 1 | 30 |
| A2L | without wind | lake patches: FLS (Heat flux that observed at LS); outside patches: FGS | 2 | 30 and 30 |
| A1LW | initial wind + geostrophic wind | Same as A1L | 1 | 30 |
| A2LW | initial wind + geostrophic wind | Same as A2L | 2 | 30 and 30 |





| A1LNG | initial wind | Same as A1L | 1 | 30 |
| A2LNG | initial wind | Same as A2L | 2 | 30 and 30 |
| A1L_C | without wind | lake patch: (SL/ST) ×FGS =(30/135)× FGS; outside patch: (SG/ST)×FGS =(105/135)×FGS | 1 | 30 |
| A2L_C | without wind | lake patches: (SL/ST)×FGS =(30/135)×FGS; outside patches: (SG/ST)×FGS =(75/135)×FGS | 2 | 30 and 30 |
| A1LW_C | initial wind + geostrophic wind | Same as A1L_C | 1 | 30 |
| A2LW_C | initial wind + geostrophic wind | Same as A2L_C | 2 | 30 and 30 |

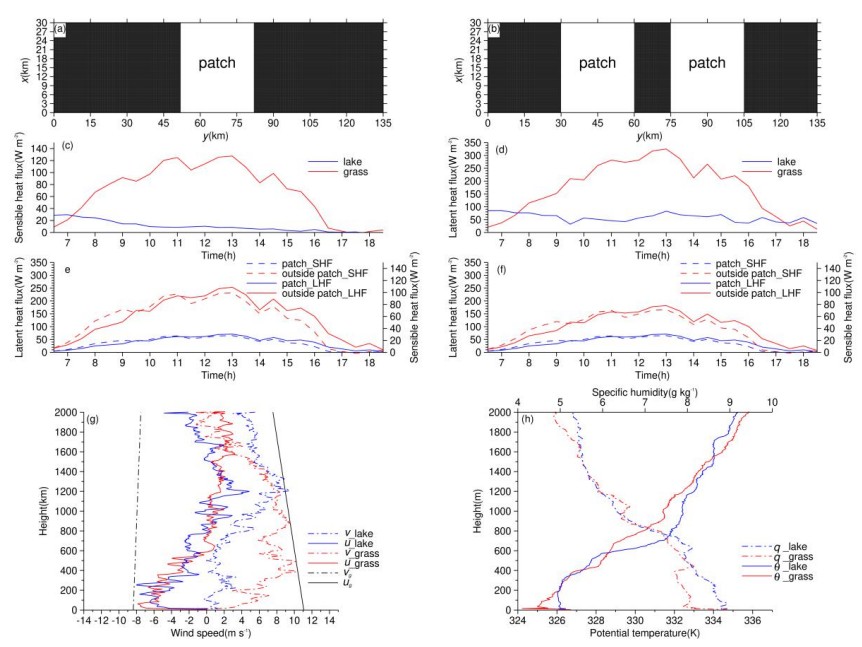





**Fig. 2.** Sketch of the heterogeneous surface (a and b), (c and d) surface sensible heat flux and
latent heat flux over the grassland (red line) and the lake (blue line) from observation. The SHF
and the LHF for runs with (e) one and (f) two lake patches and a constant heat flux. Figures 2g
and 2h show the initial profiles of the winds (solid lines for $u$, dash lines for $v$), potential
temperature (solid lines), and special humidity (dash lines) over the lake patches (blue lines) and
outside of the patches (red lines). The input geostrophic winds are also shown (black lines).
**2.2.2 statistical analysis**
According to turbulence theory, a physical quantity $\phi$ has two parts, i.e., the
horizontal average $\langle \phi \rangle$ and the turbulent fluctuation $\phi'$, and
$$\phi = \langle \phi \rangle + \phi' \tag{3}$$

This equation usually works in cases with a homogeneous surface. The variances of
velocity and the potential temperature variances ( $\sigma_v^2$, $\sigma_w^2$, $\sigma_\theta^2$ ) are calculated from
$v'$, $w'$, and $\theta'$, respectively. For a heterogeneous surface, phase-averaged analysis
helps separate the patch-induced circulations from the random turbulent motions. This
method has been applied in studies of the one-dimension and two-dimension
heterogeneities (Matthias et al., 2014; Kang and Lenschow, 2014; Shen et al., 2016)
and complex and irregular heterogeneities (Maronga and Raasch, 2013). In this study,
one-dimensional heterogeneous (in the $y$ direction) simulations were performed for
which $\phi$ can be decomposed into three parts:
$$\phi = \langle \phi \rangle + \phi_{hi} + \phi_s \tag{4}$$

Where $\langle \phi \rangle$ is the horizontal average; $\phi_{hi}$ is the heterogeneity-induced part which is
the averaged $\phi$ across the domain in the $y$ direction; and $\phi_s$ is from the background
turbulence. The variances of velocity and the potential temperature induced by the
heterogeneity ( $\left[\sigma_v^2\right]_{hi}$, $\left[\sigma_w^2\right]_{hi}$, $\left[\sigma_\theta^2\right]_{hi}$ ) are calculated from $v_{hi}$, $w_{hi}$, and $\theta_{hi}$,





respectively.
Phase-averaged analysis was also used to obtain the patch-induced component of
the turbulent fluxes. We multiplied $w$ and $\phi$ with both in the forms of Equation (4),
and derived the total vertical transport of $\phi$:
$$\overline{\langle w\phi \rangle} = \overline{\langle w \rangle \langle \phi \rangle} + \overline{\langle w_{hi}\phi_{hi} \rangle} + \overline{\langle w_s \phi_s \rangle} \tag{5}$$
Since the horizontal average vertical velocity $\langle w \rangle$ is approximately zero in the LES,
the turbulent fluxes were divided into two parts: a patch-induced circulation induced
part and a background turbulence induced part:
$$\overline{\langle w\phi \rangle} = \overline{\langle w_{hi}\phi_{hi} \rangle} + \overline{\langle w_s \phi_s \rangle} \tag{6}$$
Moreover, the total kinetic energy $e$ can be written as two parts, $e_{hi}$ and $e_s$, which
represent the contributions from the patch-induced and background turbulence:
$$e = e_{hi} + e_s \tag{7}$$
$$e_{hi} = \left( \langle u_{hi}^2 \rangle + \langle v_{hi}^2 \rangle + \langle w_{hi}^2 \rangle \right) / 2 \tag{8}$$
$$e_s = \left( \langle u_s^2 \rangle + \langle v_s^2 \rangle + \langle w_s^2 \rangle \right) / 2 \tag{9}$$
**3. Results**
**3.1. Synoptic background and wind components' distribution**
In order to investigate the existence of a daytime lake breeze (the divergent flows
over the lake surface and the downdrafts overlying it) using the ERA-Interim
reanalysis data for the two lakes area (34.8° N–35° N, 97° E–98° E; blue box in fig. 3),
the wind field, temperature field, and geopotential height field at the surface (600 hPa,
~4200 m) and at 500 hPa (~5500 m) at 12:30 LT and 18:30 LT on July 29, 2012, were
analyzed. A cyclone controlled the entire region above the surface at 12:30 LT (Fig. 3a)



and divergent flow occurred at 500 hPa at 18:30 LT (Fig. 3b). The vertical sections of
the two wind components ( *u* and *w*) were also depicted to further ascertain the
distribution of the wind field in the longitude and latitude directions. It should be
noted that downdrafts are dominant below 500 hPa in the two lakes area during the
day (Figs. 3c and 3d). As can be seen from fig. 3f, distinct divergent zonal wind (*u*)
flows existed in the two lakes area at 18:30 LT. The wind speed derived from the GPS
sounding at 12:30 LT is larger than that at 18:30 LT below 2 km (see fig. S1 in
supplement), indicating that the larger background flows covered up the divergent
wind flow at 12:30 LT. Evidently, it is difficult to directly observe the lake breeze
circulation due to the synoptic background, but the downdrafts and the divergent
zonal wind in the two lakes area demonstrate the existence of a lake breeze. In the
following sections, the turbulence characteristics over the heterogeneous underlying
surface are simulated and the effects of the patch-induced circulation are analyzed.

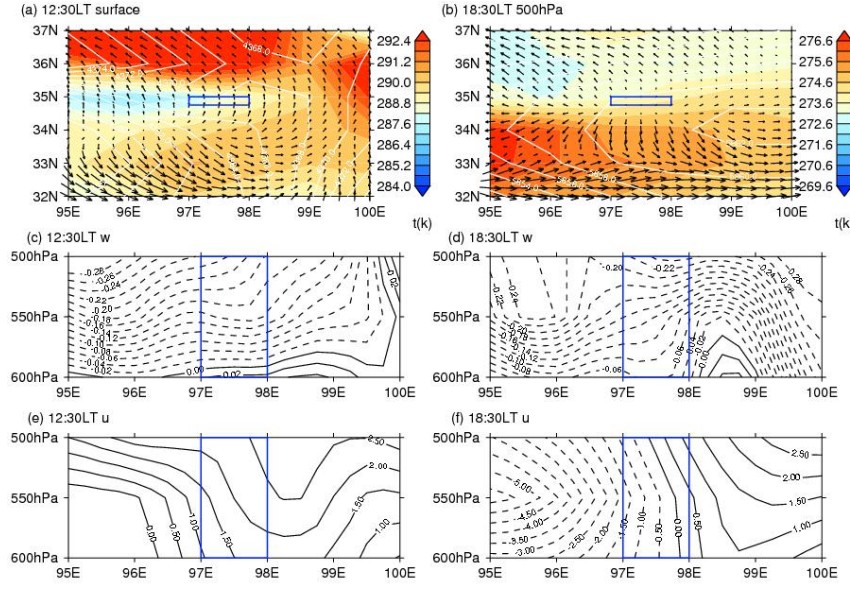



**Fig. 3.** Synoptic background on July 29, 2012. Blue boxes represent the two lakes area. (a) and (b)
show the wind field, temperature field, and geopotential height field at the surface (600 hPa,
~4200 m, 10 m wind field, 2 m temperature field; Fig. 3a) at 12:30 LT and at 500 hPa (~5500 m)
at 18:30 LT (Fig. 3b). The vertical wind ($w$, figs. 3c and 3d) and the zonal wind ($u$, figs. 3e and 3f)
below 500 hPa are also shown.
## 3.2 Effects of the underlying surfaces and background flows on the
## boundary-layer turbulence
### 3.2.1 Performance of the LEM and the height of the boundary layer over
### homogeneously heated and heterogeneously heated surfaces
In order to inspect the performance of the LEM over the heterogeneously heated
surfaces, the simulated virtual potential temperature ($\theta_v$) over the lake patch/patches
and outside were compared with the observations. In addition, by keeping the total
surface heat flux into the modeled domain constant, the profiles of the simulated
virtual potential temperature over the homogeneous and heterogeneous surfaces were
compared in order to investigate the effects of surface heterogeneity on the structure
of the boundary layer. The profiles of the kinematic heat flux for all of the runs were
used to determine the height of the boundary layer. Figures 4a and 4b compare the
simulated profiles of the virtual potential temperature over and outside of the lake
patch with the corresponding observations (solid lines) over the grassland and the lake
surfaces at different times. In order to account for the effects of the unrepresented
large-scale forcing, the simulated horizontally averaged potential temperature, water
vapor mixing ratio, and horizontal wind ($u$ and $v$) were relaxed to those observed
using the radiosondes with at a 3 h interval during the simulation (Marsham et al.,



2008; Huang et al., 2009). The time series of the kinetic energy (see fig. S2 in
supplement) indicates that the equilibration time of the model is approximately 3
hours.

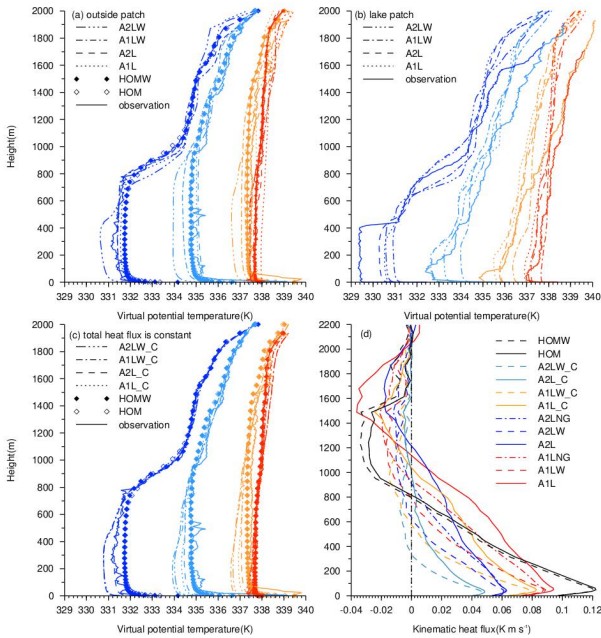

**Fig. 4.** The profile of the horizontal, averaged virtual potential temperature for the observations

and all of the runs over the lake patches (a) and outside the of the (b) lake patches. (c) same as (a)
except, for heterogeneous and homogeneous runs with constant surface heat fluxes. Legends for
(a), (b), and (c): dark and light blue represent the results at 09:30 and 12:30, respectively, and the
orange and red lines show the results at 15:30 and 18:30, respectively. (d) The kinematic heat
fluxes for all of the runs.

The observation profiles (solid lines in fig. 4a) show that the depth of the

convective boundary layer (CBL) over the grassland increases from 700 m at 09:30 to
1.1 km at 12:30 to 1.5 km at 15:30 to 1.9 km at 18:30. The inversion layer above the
CBL is completely eroded by the turbulence after 12:30. The virtual potential
temperature in the well-mixed CBL over the grassland increases approximately 7 K
from 09:30 to 18:30. The CBL over the heterogeneous surfaces with background wind
is cooler and shallower than that over the homogeneous surfaces. This may be
because the air blowing from the lake patches cools the CBL of the outside patches
that are downwind, which inhibits the development of the CBL. In addition, the model
profiles of the virtual potential temperatures over the homogeneously heated and
heterogeneously heated surfaces with no background wind have very similar
structures and are close to the sounding profiles. This is similar to the modeling
results of Liu et al. (2011). The observed virtual potential temperature over the lake
surface (solid lines in fig. 4b) shows that the CBL changes to stable stratification as
the radiation increases after sunrise, and the modeled $\theta_v$ over the patches is about
1.0 K warmer than the observed $\theta_v$. As in fig. 4a, fig. 4c also shows that the
background wind over the heterogeneous surface inhibits the growth of the CBL.

In this study, according to Sullivan et al. (1998), the height of the boundary layer ($zi$)

was determined using the minimum kinematic heat flux of the simulated results. As
can be seen, the maximum surface heat fluxes were relatively large over the
homogeneously heated surface, while smaller surface heat fluxes occurred for the case
with two lake patches. Compared to the unbalanced cases (A1L, A1LW, A2L, and
A2LW), less heat flux was introduced in the balanced cases (A1L_C, A1LW_C,
A2L_C, and A2LW_C) and lower CBLs occurred, especially with a wind field (blue
bars in fig. 5). The kinematic heat fluxes decreased to zero at higher altitude over the
heterogeneously heated surface. When the height continues to rise, the region of
negative heat flux is often called the entrainment layer, which is thicker in the cases



with background wind. The heights of the CBL indicate (Fig. 5) that the surface heat
flux anomaly may contribute to the deepening of the mixed layer, thus increasing the
CBL height. However, the shear generated by the background wind strengthens the
turbulent exchange between the entrainment layer and the free atmosphere, resulting
in an excessively thick entrainment layer, which, however, inhibits the upward
development of CBL.

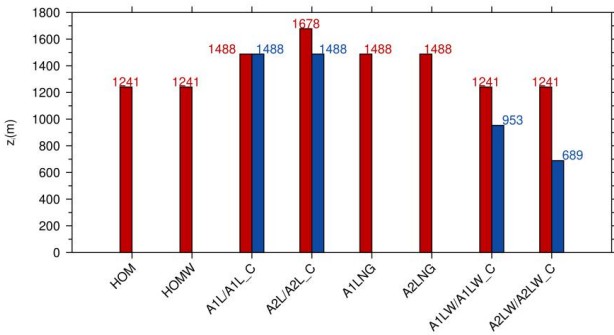

**Fig. 5.** Bar chart of the CBL height for each run marked with a concrete value. The red bars from
left to right represent runs HOM, HOMW, A1L, A2L, A1LNG, A2LNG, A1LW, and A2LW. The
blue bars from left to right represent runs A1L_C, A2L_C, A1LW_C, and A2LW_C.
**3.2.2 Effects of the surface heat flux anomalies and background winds on the**
**turbulent kinetic energy**
Local circulations will be induced by differential heating, and the turbulent kinetic
energy (TKE) determines the transport of the momentum, heat, and moisture through
the boundary layer (Tyagi and Satyanarayana, 2013). Thus, the thermal circulations
induced by the lake patches were simulated to investigate the effects of the
heterogeneous heating on the spatial distribution of the TKE.
Fig. 6 shows the vertical distribution of the TKE and the wind vectors over the
homogeneous and heterogeneous surfaces with no background wind at 15:30 LT. Over





345 the homogeneous surface (Fig. 6a), a relatively uniform TKE with a larger magnitude

346 exists within a much shallower CBL (below 0.1 $zi$), which overlies the scattered and

347 disordered wind vectors throughout the domain. Over the heterogeneous surfaces, the

348 large TKE values are distributed on both sides of the lake patches below 0.5 $zi$, and

349 the divergent winds extend to about 30 km away. In addition, a larger TKE and

350 convergent winds occurred in the upper level of the CBL (Figs. 6b, 6c). Moreover, the

351 air flow between the two lake patches led to a convergent region (updrafts in y= 0 km;

352 Fig. 6c). This is consistent with the results of Avissar and Schmidt (1998), who

353 demonstrated that turbulent eddies are randomly distributed over a homogeneous

354 surface, but the TKE exhibits two maxima near the ground surface and the top of the

355 CBL, which is in agreement with the patch-induced circulations. Overall, Figure 6

356 illustrates that the distributions of the TKE and the patch-induced circulations are

357 symmetrical on both sides of the lake patches, while the distribution is random with

358 smaller TKE values over the homogeneous surface.

359  Furthermore, the ratios of the horizontally averaged TKEs in the model domain of

360 the different runs were calculated to examine the effects of the surface anomalies and

361 the ambient winds on the TKE. As is shown in Table 2, the TKEs for the cases with

362 one or two lake patches are about twice that of the TKE of the case without patches

363 (columns 2–3), but the ambient wind leads to a reduction in the impacts of surface

364 flux heterogeneity on the TKE (columns 4–5 and 6–7). This is consistent with the

365 results of Avissar and Schmidt (1998), who reported that a weak background wind of

366 2.5 m/s is strong enough to considerably reduce the impact of the ground-surface



heterogeneity on the CBL. For the homogeneous cases, the TKE increases under the
background wind conditions due to the increase in the sheared turbulence. For the
runs with balanced surface heat fluxes (A1L_C, A2L_C, A1LW_C, A2LW_C), the
effects of the heterogeneity on the TKE are less significant, especially for the cases
with more lake patches, but the effects of the background winds on the TKE tend to
be large.

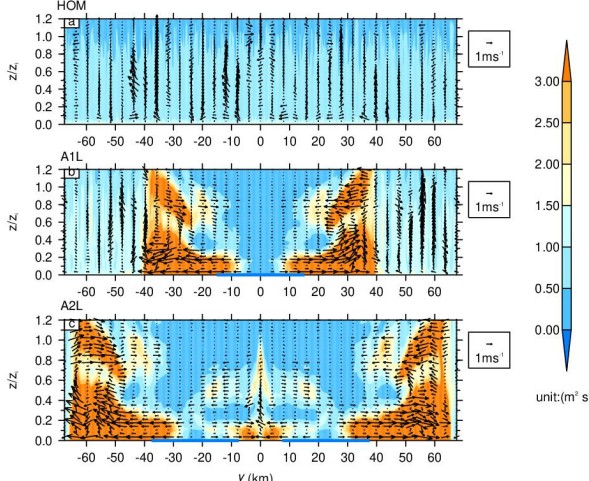

**Fig. 6.** The y-z cross sections of the TKE (contour) with superimposed wind vectors composed of
$v$ and $w$ wind over (a) homogeneously heated and (b and c) heterogeneously heated surfaces. The
blue lines on the y-axis represent the lake patches.
**Table 2**
The ratio of the TKEs of the different runs. Max, Min, and Mean stand for the maximum,
minimum, and mean ratios of the TKE in the model domain, respectively.

| Ratio of TKE | A1L/HOM | A2L/HOM | A1LW/HOMW | A2LW/HOMW | A1LW/A1L | A2LW/A2L | HOMW/HOM |
|---|---|---|---|---|---|---|---|
| Max | 3.31 | 3.30 | 1.47 | 1.42 | 1.01 | 0.88 | 2.09 |
| Min | 1.15 | 1.18 | 0.80 | 0.59 | 0.61 | 0.40 | 0.83 |
| Mean | 2.00 | 2.04 | 1.09 | 0.95 | 0.79 | 0.68 | 1.41 |
| Ratio of TKE | A1L_C/HOM | A2L_C/HOM | A1LW_C/HOMW | A2LW_C/HOMW | A1LW_C/A1L_C | A2LW_C/A2L_C | HOMW/HOM |
| Max | 2.21 | 1.44 | 1.12 | 0.74 | 1.84 | 2.17 | 2.09 |
| Min | 0.43 | 0.23 | 0.63 | 0.31 | 0.80 | 0.47 | 0.83 |





| Mean | 1.09 | 0.75 | 0.91 | 0.54 | 1.22 | 1.07 | 1.41 |
|---|---|---|---|---|---|---|---|

To further investigate the effects of the surface heat flux anomalies on the
development of turbulence, it is instructive to examine the vertical profiles of the
buoyancy and shear production terms in the TKE budget equation, which is from the
contributions of the resolved (RES) and subgrid (SGS) eddies (Figs. 7a, 7b, and 7c).

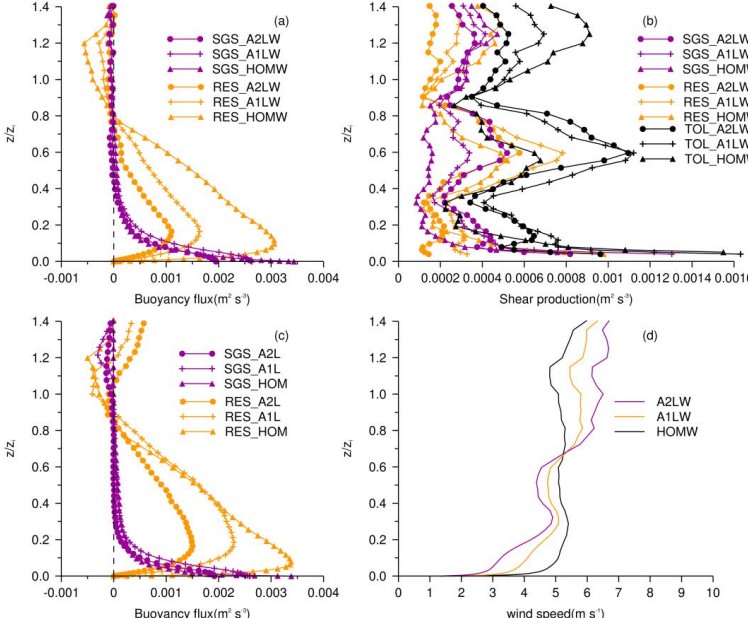

**Fig. 7.** Vertical profiles of (a) the buoyancy flux and (b) the shear production term for runs
HOMW, A1LW, and A2LW with background flows, and (c) the profiles of the buoyancy flux for
runs HOM, A1L, and A2L without background flows. (d) The simulated horizontal wind versus
height for runs HOMW, A1LW, and A2LW. The resolved and subgrid results are presented as
yellow and purple lines, respectively. The black lines in (b) are the total (resolved and subgrid
scale) shear production term.
Figures 7a and 7c show that the RES buoyancy production decreases as the number
of patches increases and the SGS buoyancy contributions are negligible, except in the
surface layer. Below 0.9 $z_i$, the larger RES shear production occurs in the case with
lake patches (Fig. 7b) and the contribution of the SGS shear production is



considerable (Fig. 7b), which is significant in the CBL for the cases with surface flux
anomalies. Thus, the total shear productions (black lines in fig. 7b) of the cases with
heterogeneous surfaces are larger. The background winds (Fig. 7d) are weaker for the
cases with lake patches below 0.65 $z_i$, but the corresponding total shear production
term is larger, which shows that the patch-induced circulations are conducive to more
shear in the CBL.
**3.2.3 Effects of the background flows on the circulations**
In order to investigate the effects of the background winds on the patch-induced
circulations, the vertical distributions of the vertical velocity and wind fields for the
runs with and without background winds were compared. In fig. 8, the patch-induced
circulations are not easy to distinguish in the cases with background winds (about
13.9 m/s above a height of 1.2 km) due to the cancellation of the local pressure
gradient by the synoptic pressure gradient, which is consistent with the results of
Crosman and Horel (2010). This also indicates that the boundary-layer convection
tends to weaken as the number of lake patches increases (the maximum updrafts are
4.8 m s$^{-1}$, 4.2 m s$^{-1}$, 3.5 m s$^{-1}$, and 3.3 m s$^{-1}$ for runs A1L, A2L, A1LW, and A2LW,
respectively). Moreover, the wind fields for the cases without geostrophic winds
exhibit divergent flows over the lake patches. As in the study of Kang and Lenschow
(2014), our study also confirms that the symmetrical patch-induced circulations and
the intensity of the convection become indistinguishable and weak under the
background flow conditions. However, the divergent flows in the lower level are still
visible when the geostrophic wind is removed (A1LNG and A2LNG in figs. 8e and



8f).

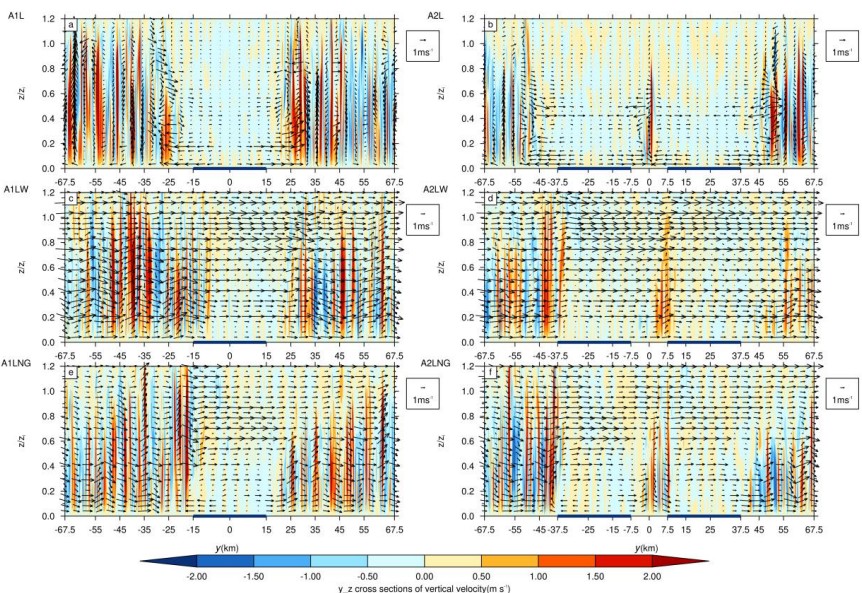

**Fig. 8.** Instantaneous y-z cross sections of the vertical velocity (m s$^{-1}$) and wind vectors above the
heterogeneous surfaces for runs (a and b) without and (c and d) with background winds, and (e
and f) with the geostrophic wind removed. The blue lines represent the lake patches.
**3.3 Effects of patch-induced circulation on the turbulent intensity**
**and heat flux**

We used the phase-averaged method to decompose the contributions of the

turbulent intensity and the heat flux from the patch-induced circulations and the
background turbulence and to quantitatively analyze the heterogeneity-induced
contribution to the turbulent intensity. For the variance of the velocity, the horizontal
(Fig. 9a) and vertical (Fig. 9b) variances induced by the heterogeneity increase as the
number of lake patches increases, and the horizontal variance is larger than the
vertical variance. However, the background flows tend to decrease both the





patch-induced (Figs. 9a, 9b) and total (Figs 9e, 9f) turbulent intensity.

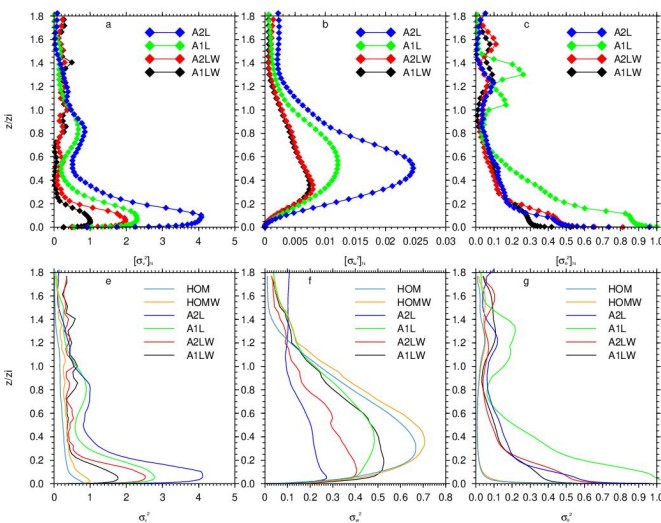

**Fig. 9.** (a, b, c) Heterogeneity-induced and (e, f, g) total dimensionless turbulence statistics for
runs HOM, HOMW, A1L, A1LW, A2L, and A2LW. Shown are the profiles of the (a, e) $v$ variance,
(b, f) $w$ variance, and (c, g) $\theta$ variance.

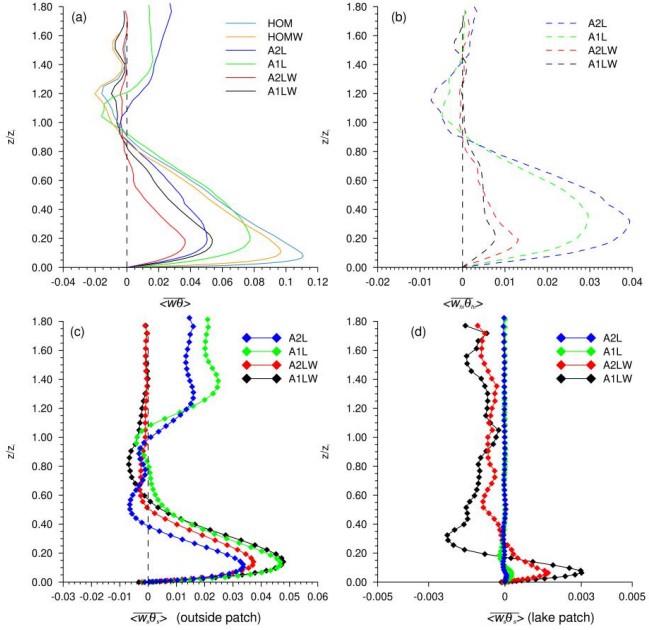

**Fig. 10.** (a) Area-averaged total turbulent heat flux (solid lines) and (b) heterogeneity-induced



turbulent heat flux (dash lines). The background turbulence (lines with diamond symbols) of heat
flux over the (c) grassland and (d) lake patches.
This also shows that a larger difference in the variances of the horizontal velocity
occurs in the surface layer and gradually decreases with height (Figs. 9a and 9e),
which means that the effects of the surface properties on the horizontal turbulence
diminish with height in the CBL. In this respect, our results are similar to those found
by Wang et al. (2011), Shao et al. (2013), and Frederik and Matthias (2018). The total
horizontal turbulent intensity is mainly from the contribution of the patch-induced
circulations and is larger than that of the homogeneous cases (Fig. 9e), which tends to
become stronger as the number of patches increases but becomes weaker as the total
vertical turbulent intensity increases (Fig. 9f, same as in the cases with balanced
surface fluxes). It should be noted that the contribution of the patch-induced
circulations to the vertical velocity variance is no more than 10% (Fig. 9b and 9f),
which implies that the background turbulence contributes more to the fluctuations in
the vertical velocity than to those in the horizontal velocity. Figures 9c and 9g show
that the patch-induced motions make the largest contribution to the variances of the
potential temperature. However, the background winds decrease the variances of the
potential temperature and decrease the impact of the surface heterogeneity on the
variances of the potential temperature.
Using the same method, we analyzed the contributions of the patch-induced and
background turbulence to the heat flux. Figure 10a shows that as the number of lake
patches increases, the area-averaged total heat flux decreases in both the mixed and

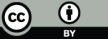



entrainment layers, and the balanced surface heat flux cases exhibit similar variations
(see fig.03 in supplement). The patch-induced transport of the heat flux increases as
the number of lake patches increases (Fig. 10b). The patch-induced motions
contribute up to 80% of the heat flux in run A2L, which has unbalanced surface fluxes
(Fig. 10b), and 61% in run A2L_C which has balanced surface fluxes (see fig. S3 in
supplement). It should also be noted that the background winds tend to decrease the
heat flux transport over the heterogeneous surfaces. As is shown in figs. 10c and 10e,
the contribution of the background turbulence to the local heat flux is larger over the
region outside of the lake patches than over the patches. We hope that the results of
our analysis of the contributions of the heterogeneity-induced circulation and
background turbulence to the turbulence intensity and the heat flux over a
heterogeneous surface will provide a basis for further studies of the local energy and
mass transport in the SRYR over the TP.
**3.4 Turbulence in the surface layer**

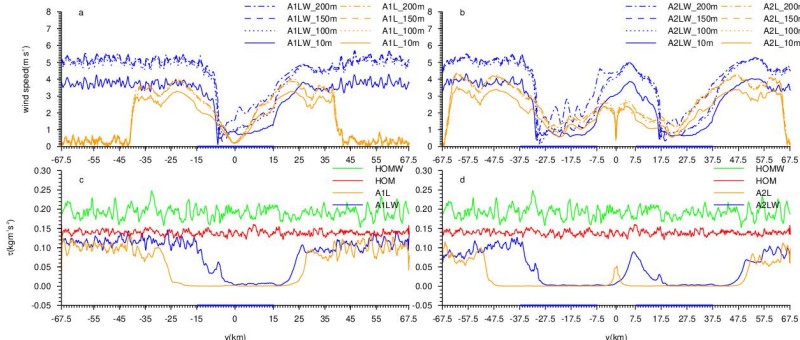

**Fig. 11.** Variations in the (a and b)wind speed and (c and d) Reynolds stress in the horizontal
direction below 200 m for the cases with (blue lines) and without (yellow lines) background flows.
The frictional velocity ($u_*$) is a critical parameter in the turbulence exchange near





the surface, and it plays an important role in the transport of momentum in the
boundary layer. Patil et al. (2016) reported that the frictional velocity increases with
increasing wind speed under lower wind speed conditions in the surface layer. Thus,
we focused on the variations in the wind speed (Figs. 11a and 11b) and Reynolds
stress (Figs. 11c and 11d) in the horizontal direction below 200 m in order to
investigate the effects of the patch-induced motions on the momentum flux in the
surface layer for various background winds. It was found that the inland extension of
the patch-induced divergent flows reached about 25 km with no background winds
(yellow curves in figs. 11a and 11b). The speed of the divergent winds increases from
the lake patches to the outside and increases with height below 200 m with and
without background winds. The wind speeds decrease rapidly (4.0 m s$^{-1}$) within 10 km
along with the wind blowing from west of the lake patches to east of the lake patch,
and then, the wind speeds increase steadily (blue curves in figs. 11a and 11b). The
changes in the surface winds and surface properties have significant effects on the
turbulent momentum flux. Figures 11c and 11d show that the transport of the
momentum flux is smaller over the heterogeneous surface. The consistent variations
in the wind speeds and the turbulent stresses illustrate that the lake patches alter the
spatial distribution of the turbulent stress, which would further affect the surface wind
speeds, especially over the land-lake boundary regions.



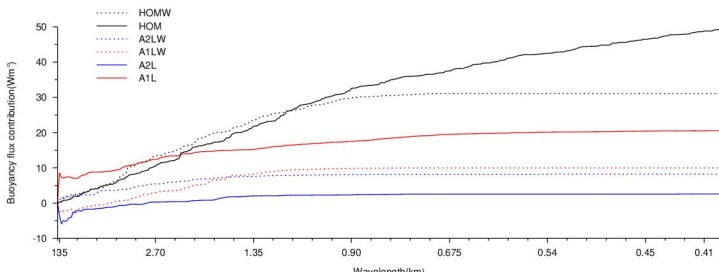

**Fig. 12.** The cumulative contribution of the buoyancy fluxes of all wavelengths (km) at a height of 50 m for runs HOM, HOMW, A1L, A2L, A1LW, and A2LW.

In order to quantify the contributions of the buoyancy fluxes, due to the different scales of the eddies, we calculated the ogives, which are the running integrals of the cospectral densities (Friehe et al., 1991), and used these values to show the cumulative contribution to the fluxes of all of the wavelengths (Brooks and Rogers, 2000). The ogive curves (Fig. 12) show that the small eddies make a significant contribution to the buoyancy fluxes over the homogeneous surfaces with no background winds (solid black line). The background wind increases the buoyancy flux for wavelengths larger than about 1.1 km and decreases it for smaller wavelengths based on a comparison of cases HOM (solid black line) and HOMW (black dotted line). The above results confirms that the heat transport is enhanced by the large eddies but is weakened by the small eddies, especially under the control of the background wind. The buoyancy flux for a wavelength larger than about 2.2 km makes a greater contribution in the case with one lake patch without background wind (solid red line). The buoyancy fluxes for wavelengths of greater than 2.7 km are transported downward for the case with two lake patches. For the case with one lake patch, the background flows tend to decrease the transport of the buoyancy flux for



larger wavelengths near the surface (red dotted line) due to the stronger horizontal
wind (Fig. 7d), and they help to transport the buoyancy fluxes downward for
wavelengths larger than 3.3 km. However, for the case with two lake patches, the
background wind causes the large eddies to transport the buoyancy fluxes upward.
Thus, increasing the number of lake patches leads to more patch-induced motions, but
this does not tend to enhance the ability of the wind to transport heat. It is concluded
that slightly more of the buoyancy flux of the case with one lake patch is transported
by the small eddies with wavelengths of less than 1.5 km compared with the case with
two lake patches and background wind conditions (red and blue dotted lines).

**3.5 The characteristics of the boundary-layer turbulence in the**

**entrainment layer**

The LES study conducted by Matthias et al. (2014) found that there is increased
entrainment from the more strongly heated surface patch cases compared to the
homogeneous cases, and the impact of the heterogeneity on entrainment vanishes due
to horizontal mixing if the mean flow is aligned perpendicular to the border between
the differentially heated patches. To investigate the effects of the thermal properties of
the heterogeneous surface and the background flows on the turbulence in the
entrainment layer, the characteristics of the heat flux in the entrainment layer were
analyzed. Our simulated results show that the downward transport of the heat flux
decreases as the number of lake patches increases in the entrainment layer for both the
wind and no wind cases (Figs. 4d and 10a), which is also true in the balanced heat
flux runs.





By comparing the maximum and minimum vertical velocities at the top of the
boundary layer (Table 3), we found that the convective intensity of the entrainment
layer in the case with two lake patches and no wind fields is stronger, but it is
weakened by the background flows. Whereas, it decreases as the number of lake
patches increases in the balanced heat flux cases (A1L_C and A2L_C), corresponding
to a smaller TKE (Table 2) and total turbulent intensity (Fig. 9f). Huang et al. (2007)
pointed out that an appropriate surface heat flux and background flows maintain the
convective roll, and our simulations demonstrate this roll-like convection (see fig. S4
in supplement), which is mainly induced by the persistence of the background
turbulence with stronger geostrophic winds of 7–11 m s$^{-1}$ (black lines in fig. 2g).
However, Maronga and Raasch (2013) found that a higher wind speed of 6 m s$^{-1}$
generates convective rolls derived from the secondary circulation over a complex
heterogeneous surface.

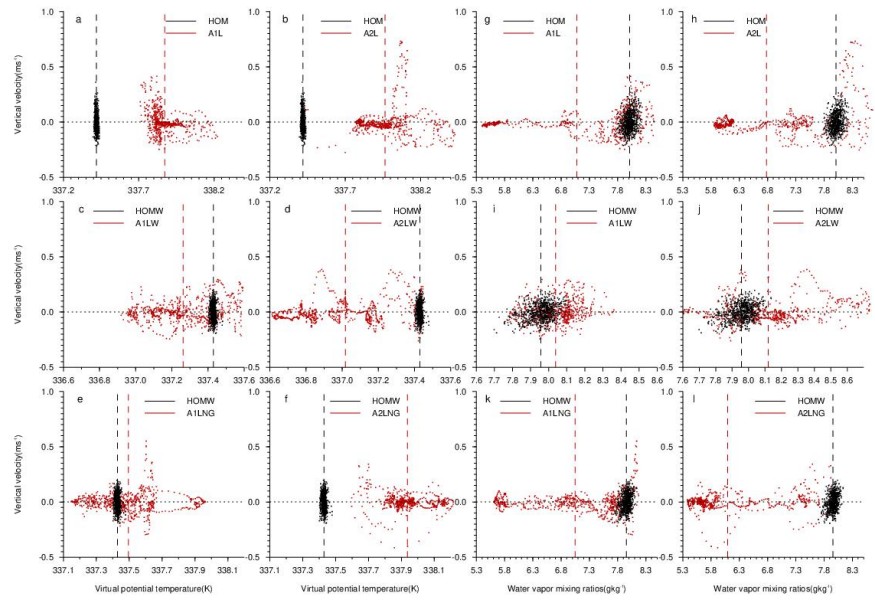



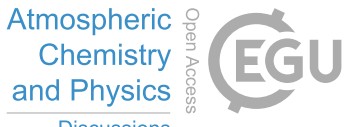

**Fig. 13.** (a, b, c, d, e, f) The joint vertical velocities and virtual potential temperatures and (g, h, i, j,
k, l) water vapor mixing ratios at the top of the CBL for the homogeneous and heterogeneous runs.
The black dotted lines represent the mean vertical velocity, and the black and red dashed lines
show the mean virtual potential temperatures and water vapor mixing ratios, respectively.
In addition, the boundary layer variables (including the vertical velocities, virtual
potential temperatures, and water vapor mixing ratios) in the entrainment layer are
also subject to the effects of the surface heterogeneity. Figure 13 shows the joint
distribution of the vertical velocities and the virtual potential temperatures, as well as
the vertical velocities and water vapor mixing ratios. Comparing to the
homogeneously heated cases, the increased downdrafts mainly occur over the lake
patches, and they carry more warm, dry air down from the free atmosphere (Figs. 13a,
13b, 13g, and 13h), which is due to the convergent airflow caused by the
patch-induced circulations at the top of the CBL. This effect is much more evident in
the case with two lake patches, but it is weakened by the gradually strengthening
background flows (except figs. 13a, 13b, 13g, and 13h). We obtained the same results
for the balanced cases. In particular, colder and moister air exists in the entrainment
layer in the cases with ambient winds (A1LW_C and A2LW_C).
**Table 3**
The maximum and minimum vertical velocities at the top of the boundary layer in cases A1LW,
A2LW, A1LW_C, A2LW_C, A1LNG, A2LNG, A1L, A2L and HOM, HOMW.

| Case | A1LW | A2LW | A1L | A2L | HOM | HOMW |
|---|---|---|---|---|---|---|
| W(max) | 4.01 | 3.54 | 5.42 | 5.55 | 4.37 | 4.48 |
| W(min) | −2.29 | −1.98 | −2.40 | −3.12 | −2.15 | −2.50 |
| Case | A1LW_C | A2LW_C | A1L_C | A2L_C | A1LNG | A2LNG |
| W(max) | 2.97 | 2.06 | 4.26 | 2.46 | 3.90 | 4.46 |
| W(min) | −2.03 | −1.25 | −1.91 | −1.27 | −2.23 | −1.97 |





## 4  Summary and discussion


The downdrafts and divergent zonal wind in the two lakes area obtained from the
ERA-Interim reanalysis data indicate the existence of a lake breeze in the SRYR. Ten
runs of the 1D strip-like distribution of the surface heat flux and two homogeneously
heated runs based on the observations made during the summer of 2012 in the
Ngoring Lake Basin were conducted in order to investigate the effects of the
patch-induced circulations on the boundary-layer turbulence and its energy transport
at the lake-air and grass-air interfaces, and the influence of the background flows also
be considered.
The thermodynamic heterogeneity of the surface is conducive to deepening the
mixed layer, thus increasing the CBL height and enhancing the TKE when there are
no background flows. The background flows bring shear, resulting in an excessively
thick entrainment layer, which inhibits the growth of CBL and reduce the effects of
the heterogeneously heated surface on the TKE. The distribution of the TKE over the
heterogeneously heated surface is consistent with the patch-induced circulations
described by Avissar and Schmidt (1998). In addition, the surface heat anomaly and
background winds have similar effects on the CBL in the cases with a balanced
surface heat flux, but the enhanced effects on the TKE are far lower in the cases with
an unbalanced surface heat flux. Thus, it is more beneficial to consider the ambient
winds. By analyzing the buoyancy and shear production terms in the TKE budget
equation and separating the contribution of the resolvable-scale (RES) and
subgrid-scale (SGS) eddies, we found that the contributions of the wind shear to the



TKE from the SGS eddies are considerable in the CBL (below $0.9zi$) over a
heterogeneously heated surface. The total shear production term is larger below $0.65zi$
in the heterogeneously heated cases with weaker background winds, demonstrating
that the patch-induced circulations are conducive to producing more shear in the CBL.
We obtained the same conclusion as Kang and Lenschow (2014), that is, the
patch-induced circulations become indistinguishable under background flows
conditions, and the ambient winds also weaken the convective intensity.

Then, we conducted a phase-averaged analysis to separate the contributions of the

turbulent intensity and the transport of the total heat flux from those of the
patch-induced circulations and the background turbulence field. The patch-induced
turbulent intensity increases with increasing lake patches. It mainly contributes to the
horizontal turbulent intensity and the potential temperature variance, while it
contributes no more than 10% to the vertical turbulent intensity, of which the
background turbulence contributes the most. The ambient winds weaken the
patch-induced and horizontal turbulent intensities but strengthen the vertical turbulent
intensity. The contribution of the patch-induced heat flux was up to 80% in the
unbalanced cases and 60% in the balanced cases. The background turbulence made a
larger contribution to the heat flux over the area outside of the patches, which have a
stronger surface heat flux than that over the lake patches. The background flows also
inhibit the transport of the heat flux.

To understand the effects of the patch-induced motions on the momentum flux in

the surface layer under various background wind conditions, we focused on the



variations in the wind speed and the Reynolds stress in the horizontal direction below
200 m. Without ambient winds, the inland extent of the patch-induced flows was
about 25 km. When the background winds flowed into the lake patches, they
decreased by 4.0 m s$^{-1}$ within about 10 km and increased steadily when flowing out of
the patches. The synchronized variations in the wind speed and momentum flux in the
horizontal direction illustrate that the lake patches alter the spatial distribution of the
turbulent stress, which further affects the surface wind speeds, especially over the
land-lake boundary regions. We also analyzed the cumulative contribution of eddies
with different scales to the buoyancy flux near the surface. It was found that without
background flows, the buoyancy flux is transmitted upward by the eddies with larger
wavelengths for the case with one lake patch; while there is a negative buoyancy flux
in the case with two lake patches. Thus, increasing the number of lake patches leads
to more patch-induced motions, which do not tend to enhance the heat transport
ability. The background flows promote the opposite results.

In the entrainment layer, in contrast to Matthias et al. (2014) who found that the

entrainment increased for the stronger heated surface patch cases compared to the
homogeneous case, we found that the entrainment flux decrease as the number of lake
patches increases. For the unbalanced cases, the convective intensity increases as the
number of lake patches increases, but the background flows weaken it. For the
balanced cases, the convective intensity weakens as the number of lake patches
increases, corresponding to a smaller TKE and total turbulent intensity. In this study,
whether the convective rolls persist mainly depended on the background turbulence



field with a higher geostrophic wind of 7–11 m s$^{-1}$, while Maronga and Raasch (2013)
reported a higher wind speed of 6 m s$^{-1}$. As the number of lake patches increases, the
increased downdrafts are mainly located over the lake patches, and they carry more
warm, dry air down from the free atmosphere in both the balanced and unbalanced
cases. The background winds weaken this effect even when there is cooler, moister air
in the entrainment layer in the balanced cases.

Our study provides ideal simulations of the boundary-layer turbulence over the

heterogeneously heated surface in the SRYR. It mainly focused on the influences of
the heterogeneous distribution of the surface heat flux and the background winds. In
the future, we plan to conduct further research that will take into consideration the
topography and additional physical processes to provide a reference for the study of
the energy and water exchange processes over the complex surface of the SRYR.
**Acknowledgments**

This research was supported by the second Tibetan Plateau Scientific Expedition

and Research Program (STEP, 2019QZKK0604), the National Natural Science
Foundation of China (NSFC) (91837208, 41775013). Thanks to the Zoige Plateau
Wetland Ecosystem Research Station, Chinese Academy of Sciences for the field
observation data, it supported by the Science and Technology Plan of Gansu Province
(20JR10RA070).

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
