# Peer review of "Large eddy simulation of boundary-layer turbulence over the heterogeneous surface in the Source Region of the Yellow River"

_Atmospheric Chemistry and Physics, 2021_

## Referee Comment (RC3)

August 10, 2021
Subject: Review of manuscript ACP-2021-325

Dear Authors

The authors employed Large Eddy Simulations (LES) to study the impacts of heterogeneous heat flux surface on flow turbulent characteristics in this study. From my point of view, the authors were able to handle this task satisfactorily and this paper is worth publishing on ACP. My decision is "Accept with minor revision".

Furthermore, I have highlighted some points of the article to be reviewed as well as I have made some comments to improve the manuscript.

**General comments**

(a) The figures need resolution improvement.

(b) The authors used many acronyms to refer to the different simulations. In my opinion, this can confuse the reader in a first moment and it makes the reading slower. On the other hand, I understand the necessity to use it. My suggestion is to explain the logic behind the character choice and try to simplify the acronyms, i.e. shorter acronyms. Furthermore, avoid using non-alphanumeric symbols like underscore.

**Specific comments**

(1) Page 7, lines 135-140 – The authors employed ERA-Interim data, with 1x1 degree resolution, to describe the flow synoptic features. Why did they chose this specific data set instead of another one with better time and spatial resolution as ERA5? A short explanation about this choice would be appreciated.

(2) Page 7, line 151 – The authors used 200-m of spatial resolution on their simulations. Is it an appropriate resolution for LES regarding the turbulence gray zone? In some papers on literature, this order of resolution size is called Very Large Eddy Simulations (VLES). Maybe the authors could clarify it better on the text. Furthermore, how about time resolution (time step) for these simulations? Please include a brief comment about it on the text.

(3) Page 7, line 152 – The authors did a vertical grid spacing description at that point. I suggest a more precise description including:

   (i) how many vertical levels were used on simulations setup;

   (ii) how they were stretched;

(4) Page 8, lines 158-161 – The authors indicated the initial conditions for the simulations. Could they mention figure 2 here? A simple indication as: "These conditions will be detailed on figure 2" would be enough.

(5) Page 11, figure 2: The initial conditions obtained from upper air sounding present a non-smooth shape with a pronounced vertical variation (mainly for the wind components), which is expected for high-resolution measurement. My question is: were they used exactly as it was showed on figure 2 as initial condition or were they smoothed to

accomplish it? If so, please, show the initial condition smoothed profile on figure 2 as well.

(6) Page 14, figure 3 – About this figure, I have the following concern/suggestions:

(i) It is not clear to me why the authors mixed different vertical levels to compare temperature and other variables at different times. Could you explain it better? For me, it makes more sense to compare same levels at different times.

(ii) The lines for geopotential height are not clear on these plots as well as its labels (values). Could you improve that?

(iii) Each wind vector seems to be plotted at 0.25 degree. Is it an interpolation for the ERA Interim 1x1 degree resolution data? If so, please remark it. Again, if the resolution for the synoptic wind field is an important feature, it seems to me that another reanalysis dataset would be more interesting for this work.

(7) Page 21, figure 7 – About this figure I have the following concern/suggestions:

(i) What day time are these profiles related to? It was not clear on the text.

(ii) I suggest using the designation "Buoyancy production/destruction" instead "Buoyancy flux" to refer to the turbulent kinetic energy (TKE) budget equation term to avoid any misinterpretation. Furthermore, "Buoyancy flux" is not precise to describe it on a physical sense.

(iii) The buoyancy production/destruction profiles showed a similar behavior for wind/no wind simulations in a homogeneous surface. However, for heterogeneous surface simulations, it is clear that resolved heat flux reaches a lower maximum and decreases differently from pure-convection (no wind) simulation. Could you briefly explain it on the text?

(iv) I suggest a new figure, like figure 6, to describe the effect of buoyancy TKE budget term. It could be included on the main paper or on the supplementary documentation. It would clarify the interaction of lake patches with atmospheric flow and how it impacts the TKE balance.

(v) Is the wind shear the source of SGS shear production peak at $z/z_i = 0.6$ or is there an unusual feature on the momentum flux profile? Again, a plot with the momentum flux profile (resolved and SGS) could be presented on the supplementary documentation to clarify it.

(vi) The wind profiles presented an interesting feature on heterogeneous surface simulations. In the homogeneous case, the wind profile seems to be log-linear close to surface and showed a clear mixed layer above it. For the lake simulations, the wind profiles exhibit a feature similar to a stable boundary layer, with a maximum local wind. It is an interesting feature that could be better discussed on the text. I suggest plotting the potential temperature profile associated to these wind profiles to better understand the PBL vertical structure at this time. One could say that an internal boundary layer process would be occurring here. Furthermore, I suggest plotting a log-linear law and the geostrofic wind components with wind profiles to better visualize and discuss these wind profiles.

(vii) Regarding the possible internal boundary layer formation, an extra plot for potential temperature, similar to figure 6, could be made.

(8) Page 22, line 408 – I wonder if the weaker updrafts could explain the buoyancy TKE budget term features highlighted previously. What do the authors think about that? If these two characteristics are related, please detail it on the text.

(9) Page 23, figure 8 – I think it would be interesting to add an extra plot here with the homogeneous cases. It helps to evaluate the lake patch impact on the local circulation.

(10) Page 432, figure 10 – What time is it on the simulation? Is it on the same time of wind profiles from figure 7? I am asking it because the negative heat flux on the wind-simulations, above the lake patch, could be decreasing the turbulent viscosity and increasing the wind speed consequently. What do the authors think about that? It is important to note that the minimum flux value (negative) is in a magnitude so strong as it is close the surface. Furthermore, it happens around the same height of local maximum wind. I suggest plotting the potential temperature associated to these heat fluxes to better understand it.

(11) Page 26, figure 11 – This figure shows a clear transition between the land-lake PBL. I would like to see the wind speed of homogeneous simulation to compare it with the heterogeneous ones, as well as a comparison for potential temperature and heat flux. It could be interesting to understand a possible internal boundary layer formation.

---

## Author Comment (AC1)

**Response to anonymous Referee #1**

We would like to thank the reviewer for taking the time to make detailed and useful comments. Details of the changes made are now given in the supplement in reply to the comments made.

**General comments:**

-Add Liu et al. (2018, 2020) into Introduction.

Liu, R., Liu, S.M., Yang, X.F., Lu, H., Pan, X.D., Xu, Z.W., Ma, Y.F., Xu, T.R., 2018. Wind dynamics over a highly heterogeneous oasis area: an experimental and numerical study. J. Geophys. Res. 123, 8418–8440.

Liu, R., Sogachev, A., Yang, X., Liu, S., Xu, T., Zhang, J. 2020. Investigating microclimate effects in an oasis-desert interaction zone. Agricultural and Forest Meteorology, 290, 107992.

Thanks for your comment. We have added Liu et al. (2018, 2020) into Introduction: "In addition, several studies have examined the effects of surface heterogeneity on different levels of background winds (Shen and Leclerc, 1995; Liu et al., 2020) and the direction relative to the orientation of the heterogeneity (Wang et al., 2011; Kang and Lenschow, 2014), as well as the spatial and temporal variations of the wind fields (Liu et al., 2018)."

-Line 82: Should describe specifically what kind of model is.

Thanks for your comment. We have described specifically what kind of model in this sentence: "Biermann et al. (2014) and Wang et al. (2015) discovered that the turbulent flux of Lake Nam co, which is surrounded by wet grasslands, is actually

very considerable but often underestimated by a hydrodynamic multilayer model from Foken (1979 and 1984)."

-Lines 151-152: "horizontal grid-spacing of 200 m. A vertically stretched grid with a minimum spacing of 1.1 m was utilized in the surface layer and a maximum of 64.8 m above 2000 m", 200:1.1, very large ratio. Is it ok for simulation?

The irregular grid spacing in vertical direction with very fine grid spacing in the surface layer has been used to simulate the turbulent fluxes for the small eddies. According to the other reviewer's comment, we have illustrated that horizontal resolution of 200 m in our simulation lies in the near gray zone during the early CBL development (12:30), but is an appropriate resolution for the time of 15:30. Please find the detailed replies in the response for the reviewer #2 of RC3. Moreover, the similar grid spacing ratio for the LEM has been used by Huang et al. (2010) to simulate the effects of surface heat flux anomalies on the formation of deep boundary layer over the Sahara dessert.

*Huang Q, Marsham J H , Parker D J, et al. Simulations of the effects of surface heat flux anomalies on stratification, convective growth, and vertical transport within the Saharan boundary layer[J]. Journal of Geophysical Research Atmospheres, 2010, 115.*

-Line 171: What's the real size of Ngoring Lake and Gyaring Lake? Does 30 km width reflect the real lake?

The spreads of Ngoring Lake and Gyaring Lake are 610 $km^2$ and 520 $km^2$, respectively. Ngoring Lake is the biggest lake in the Source Region of the Yellow River and ranges about 30 km from the south to north (west to east) according to Wen et al. (2015). The 30 km×30 km×6 km of lake domain size can represent Ngoring Lake and Gyaring Lake.

*Wen L J, Lv S H, Li Z G, et al. 2015. Impacts of the two biggest lakes on local temperature and precipitation in the Yellow River source region of the Tibetan Plateau [J]. Adv. Meteor., 2015: 248031. doi:10.1155/2015/248031.*

-Line 249: Please specify the time period of the daytime.

Thanks for your comment. We have corrected the sentence as: "In order to investigate the existence of a daytime (6:30-18:30 LT) lake breeze......"

-Line 290: Please explain what the "h" means is.

Thanks for your comment. Here "3 h" means three hours. We have changed "······ using the radiosonde with at a 3 h interval······" to "······using the radiosonde with a 3 h (hour) interval······"

-Lines 306-308: Please cite refs that can support this hypothesis.

Thanks for your suggestion. We have modified the hypothesis and cited the reference as: "This may be because the background wind weakens the boundary-layer convection, which inhibits the development of the CBL (Huang et al., 2009)."

-Figure 5: The heights of A1L and A1L_C is the same, why? The vertical coordinate may be $z_i$?

In our study, the height of the boundary layer ($z_i$) was determined using the minimum kinematic heat flux of the simulated results according to Sullivan et al. (1998). There is a very similar surface heat flux (only about 0.1 $W \cdot m^{-2}$ difference between the cases A1L and A1L_C) that driven the development of the CBL for the runs A1L and A1l_C, which determines the same CBL height for the both cases. In addition, we have changed the vertical coordinate according to your comment.

-Lines 368 – 372: Why the effects of the heterogeneity on the TKE for the runs with balanced surface heat fluxes are less significant? Have other studies have the same phenomenon?

For unbalanced surface heat fluxes simulations (A1L, A2L, A1LW, A2LW), runs with one or two lakes were initialized using the surface heat fluxes measured at LS (the lake station) for the lake patches and the heat fluxes at GS (the grassland station) for the outside patches. For balanced surface heat fluxes runs (A1L_C, A2L_C, A1LW_C, A2LW_C), the surface heat fluxes measured at GS are allocated according to the proportion of the square of lake patches and outside patches to the model domain. Based on the above approach, surface heat fluxes for the unbalanced runs are larger (about 0.1-15 $W \cdot m^{-2}$ for the sensible heat flux and 0.3-38 $W \cdot m^{-2}$ for the latent heat flux) than that for the balanced runs. It results the smaller TKE for the balanced runs. In order to present clearly, we have modified the sentence as: "For the runs with balanced surface heat fluxes (A1L_C, A2L_C, A1LW_C, A2LW_C), the effects of the heterogeneity on the TKE are less significant due to the relative smaller surface heat fluxes..." The similar simulation settings are found in the LEM study by Huang et al. (2010). It was found that effects of the surface heat flux anomaly on the CBL changes tended to be small for runs used balanced surface heat fluxes (see Fig. 4 from Huang et al. (2010)).

*Huang, Q., Marsham, J. H., Parker, D. J., Tian, W. S., and Grams, C. M.: Simulations of the effects of surface heat flux anomalies on stratification, convective growth, and vertical transport within the Saharan boundary layer, J. Geophys. Res., 115, D05201. doi:10.1029/2009JD012689, 2010.*

Lines 399 – 415: Discuss the result with Zhou et al.(2018) and Liu et al. (2020).

Zhou, Y., Li. D., Liu, H. and Li, X.: Diurnal variations of the flux imbalance over homogeneous and heterogeneous landscapes. Boundary-Layer Meteorology, 168:417–442. https://doi.org/10.1007/s10546-018-0358-2, 2018.

Liu, R., Sogachev, A., Yang, X., Liu, S., Xu, T., Zhang, J. 2020. Investigating microclimate effects in an oasis-desert interaction zone. Agricultural and Forest Meteorology, 290, 107992.

Thanks for your comment. Here we added the following statement on line 410: "Zhou et al. (2018) and Liu et al. (2020) investigated desert-oasis microclimate effects by simulations and found background wind has crucial effects on the thermal heterogeneous system. They showed the similar results that the circulations are more pronounced between the hot and cold patches without background wind. As background wind increased, the local circulation is gradually weakened and eventually replaced by horizontal flows over the oasis-desert system."

-Section 4: What's the difference between ambient wind, background wind and background flow?

In our text, the ambient wind, background wind and background flow are the same thing, which refer to the runs with both the initial wind and geostrophic wind. We have used the expression - "background wind" throughout the text to avoid confusing.

**Specific comments:**

-Line 48: The sentence "…, which has improved our understanding of the transfer and spatial and temporal variability of the turbulence" is a bit redundant and it is unclear exactly what point you are making. I would suggest rephrasing.

Thanks for your suggestion. We have rewritten this sentence: "Then turbulence over heterogeneous surfaces was investigated through field campaigns (Wang et al., 2016; Zhao et al., 2018) and numerical simulations (Shao et al., 2013; Liu et al., 2011) in the past few decades, which help us better understand the interactions between the surface and atmosphere."

-Lines 128 - 133: I would suggest rephrasing the sentence.

Thanks for your suggestion. This sentence was rephrased like this: "The GPS radiosonde data is obtained from the field campaign on July 29, 2012, at 30 m west of Lake Ngoring (near the gradient tower station, TS) and Madoi station (MD) located

30 km east of the lake (34.918°N, 98.216°E, 4279 m AMSL). The eddy covariance data for Lake Station (LS) above the northwest of the lake (35.026°N, 97.652°E) and Grassland Station (GS) (34.913°N, 97.553°E) at 1.5 km west of the lake shore were used."

-Line 135: There are two "and", please rephrase the sentence. The same as Line 217, Line 225 etc. Please check throughout the manuscript.

We have rephrased the sentence at Line 135: "The synoptic background at 500 hPa and the distribution of the wind components..." The sentence at Line 217 have rephrased: "the horizontal average $\langle \phi \rangle$ and the turbulent fluctuation $\phi^{'}$, so..." The sentence at Line 225 have rephrased: "..., as well as the complex and irregular heterogeneities (Maronga and Raasch, 2013)."

-Line 141: When was the Landsat image acquired?

The landsat image was acquired on 21th August, 2014. We have added the time of this image in the caption of Fig. 1.

-Line 150: Please specific the "135 km×30 km×6 km" means: Length? Width? Height?

We have rewritten this sentence: "The domain size was 135 km×30 km×6 km in the $y$, $x$ and $z$ direction, respectively, with a horizontal grid-spacing of 200 m."

-Figure 3: The legend should be "T(K)" in figs 3a and 3b. What's are the units of w and u in figs 3c and 3d?

We have modified the legend in Figs. 3a and 3b, and added the units (Pa s-1) and (m s-1) of $w$ and $u$ in Figs. 3c and 3d.

-Line 456: "fig.03" should be "Fig.S3". The same as line 459. Please check "fig" or "Fig" throughout the manuscript and make them unified.

We have changed the "fig. 03" into "Fig. S3". We have used the unified "Fig" throughout the manuscript.

Technical corrections:

-Line 32: Miss interpunction between "m" and "s$^{-1}$".

We have modified the unit as: "m·s$^{-1}$".

-Line 35: Change "it" to "them".

We have changed "it" to "them".

-Line 90: Delete "," after "heat".

We have deleted "," after "heat".

-Line 119: Should be the entrainment layers of PBL.

We have changed the sentence: "...and the entrainment layers of PBL were investigated, too."

-Line 122: "data" should be "Data".

We have changed "data" to "Data".

-Line 248: "distribution" should be "distributions".

We have changed "distribution" to "distributions".

-Line 474: Delete increasing.

We have deleted the "increasing" .

---

## Author Comment (AC2)

**Response to anonymous Referee #2**

We would like to thank the reviewer for taking the time to make detailed and helpful comments. The comments have been carefully addressed and our replies are summarized below.

**General comments**

(a)The figures need resolution improvement.

Thanks for the comments. We have improved the resolutions of all figures in the revised version.

(b)The authors used many acronyms to refer to the different simulations. In my opinion, this can confuse the reader in a first moment and it makes the reading slower. On the other hand, I understand the necessity to use it. My suggestion is to explain the logic behind the character choice and try to simplify the acronyms, i.e. shorter acronyms. Furthermore, avoid using non-alphanumeric symbols like underscore.

Thanks for the good suggestion. We have given the logical description of the characters in the title of Table 1, as: "Parameters for the 3D simulations over the homogeneous surface (HOM) and heterogeneous surface with surface heat flux anomalies (A) under different conditions: with one (1L) or two (2L) lake patches, with initial wind and geostrophic wind (W), without geostrophic wind (NG), with the constant surface heat flux (C)". We also have changed the test names using the constant surface heat flux of A1L_C, A2L_C, A1LW_C, and A2LW_C into A1LC, A2LC, A1LWC, and A2LWC in the text.

**Specific comments**

(1) Page 7, lines 135-140 – The authors employed ERA-Interim data, with 1x1 degree resolution, to describe the flow synoptic features. Why did they chose this specific data set instead of another one with better time and spatial resolution as ERA5? A short explanation about this choice would be appreciated.

Thanks for your suggestions. Sorry, we described the resolution of the reanalysis data not clearly in the paper. We used the ERA-Interim Reanalysis Data with a 0.25° × 0.25° resolution for the zonal and vertical winds in Fig. 3, and 1° × 1° resolution for temperatures and geopotential heights. We paid more attention to the synoptic wind field and the circulations induced by the surface heat flux anomaly. We have replotted Fig. 3 using ERA-Interim reanalysis data with a 0.25° × 0.25° resolution for all variables. We have changed in the text: "...using the ERA-Interim Reanalysis Data with a 1° × 1° resolution..." to "...using the ERA-Interim Reanalysis Data with a 0.25° × 0.25° resolution..."

(2) Page 7, line 151 – The authors used 200-m of spatial resolution on their simulations. Is it an appropriate resolution for LES regarding the turbulence gray zone? In some papers on literature, this order of resolution size is called Very Large Eddy Simulations (VLES). Maybe the authors could clarify it better on the text. Furthermore, how about time resolution (time step) for these simulations? Please include a brief comment about it on the text.

Thanks for your comments. A large horizontal domain (135 km × 30 km) is used to include possible mesoscale circulation due to the surface heat flux anomaly in this study. Considering the high computational cost, we employed a grid spacing of 200 m for the LES simulations. However, previous studies about the turbulence gray zone confirm the spatial resolution of 200 m in our study is appropriate.

Honnert et al. (2011) defined a dimensionless mesh size $\Delta x/(z_i+z_c)$ to quantify the resolved and subgrid parts of the turbulence at different scales of any free convective boundary layer, where $z_i$ and $z_c$ are the ABL height and the depth of the shallow cloud

layer, respectively. Honnert et al. (2011) found that the resolved and subgrid TKE are equal for $\Delta x/(z_i+z_c) = 0.2$. In our study, the CBL height reaches to about 700 m at 09:30 and up to 1900 m at 18:30 (Fig. 4 in the paper). Clouds developed from a cloud base approximately 1000 m at about 12:30 (Fig. R1).

[Figure]

Fig. R1 The heights of cloud base for different runs.

[Figure]

Fig. R2 The partition of the LES, near gray-zone, gray-zone and mesoscale (Shown in Fig. 4 from Honnert et al. (2020))

The calculated dimensionless mesh size is about 0.12 at 12:30, and about 0.08 at 15:30, which indicates the resolved TKE is larger than the subgrid part, especially for the time of 15:30 (Fig. R2). Accoording to Honnert et al (2020), the CBL gray zone is roughly at 200 m < $\Delta x$ < 2 km when LES converging simulations is achieved at $\Delta x \sim$ 20 m with taking $z_i$ = 1000 m. It illustrates that the horizontal resolution of 200 m in our simulation lies in the near gray zone during the early CBL development (12:30),but is an appropriate resolution for the time of 15:30.

Some LES studies have used horizontal grid spacing of larger than 200 m to investigate the CBL turbulence over the heterogeneous surface (Huang et al. 2010; Rai et al. 2016; Xu et al. 2018). For example, Huang et al. (2010) used the Met Office Large Eddy Model with grid spacings of 200 m to simulate the effects of surface heat flux anomalies on the formation of deep boundary layer over the Sahara dessert.

We have clarified the resolution choice: "According to Honnert et al. (2011) and Honnert et al. (2020), the horizontal resolution of 200 m is reasonable in this LEM study."

The time step of 0.01s is applied for all simulations in this paper. According to your suggestion, a brief comment is added: "The time step is 0.01s for all simulations."

Honnert, R., Masson, V., & Couvreux, F. (2011). A diagnostic for evaluating the representation of turbulence in atmospheric models at the kilometric scale. J. Atmos. Sci., 68 , 3112-3131.
Honnert R , Efstathiou G A , Beare R J, et al. The Atmospheric Boundary Layer and the "Gray Zone" of Turbulence: A Critical Review[J]. Journal of Geophysical Research: Atmospheres, 2020, 125.
Huang Q, Marsham J H , Parker D J, et al. Simulations of the effects of surface heat flux anomalies on stratification, convective growth, and vertical transport within the Saharan boundary layer[J]. Journal of Geophysical Research Atmospheres, 2010, 115.
Rai R K , Berg L K , Kosovi B, et al. Comparison of Measured and Numerically Simulated Turbulence Statistics in a Convective Boundary Layer Over Complex Terrain[J]. Boundary-Layer Meteorology, 2016, 163(1):1-21.
Xu H, Wang M , Wang Y, et al. Performance of WRF Large Eddy Simulations in Modeling the Convective Boundary Layer over the Taklimakan Desert, China[J]. Journal of Meteorological Research, 2018.

(3) Page 7, line 152 – The authors did a vertical grid spacing description at that point. I suggest a more precise description including:

(i) how many vertical levels were used on simulations setup;

(ii) how they were stretched;

Thanks for your suggestions. Seventy-four (74) levels were set up in the vertical direction. We have revised the statement as: "There were 74 levels in the vertical direction, with a vertically stretched grid having a minimum spacing of 1.1 m in the surface layer and a maximum of 64.8 m above 2000 m."

(4) Page 8, lines 158-161 – The authors indicated the initial conditions for the simulations. Could they mention figure 2 here? A simple indication as: "These conditions will be detailed on figure 2" would be enough

Thanks for your reminding. We have added the description as your suggestion: "These conditions will be detailed on Fig. 2."

(5) Page 11, figure 2: The initial conditions obtained from upper air sounding present a non-smooth shape with a pronounced vertical variation (mainly for the wind components), which is expected for high-resolution measurement. My question is: were they used exactly as it was showed on figure 2 as initial condition or were they smoothed to accomplish it? If so, please, show the initial condition smoothed profile on figure 2 as well.

Thanks for your comments. Sorry about the inaccurate description of the initial profiles. The initial profiles shown on Figs. 2g and 2h are the radiosonde initial profiles, which were interpolated to the model grid as the LEM initial profiles. We have revised the caption of Fig. 2 and added the model initial profiles in Fig. 2 as well.

[Figure]

Fig. 2. Sketch of the heterogeneous surface (a and b), (c and d) surface sensible heat flux and latent heat flux over the grassland (red line) and the lake (blue line) from observation. The SHF and the LHF for runs with (e) one and (f) two lake patches and a constant heat flux. Figs. 2g and 2h show the initial profiles of the winds (solid lines for $u$, dash lines for $v$), potential temperature (solid lines), and special humidity (dash lines) over the lake patches (blue lines) and patches outside (red lines) in LEM. The input geostrophic winds are also shown (black lines)

(6) Page 14, figure 3 - About this figure, I have the following concern/suggestions:

(i) It is not clear to me why the authors mixed different vertical levels to compare temperature and other variables at different times. Could you explain it better? For me, it makes more sense to compare same levels at different times.

(ii) The lines for geopotential height are not clear on these plots as well as its labels (values). Could you improve that?

(iii) Each wind vector seems to be plotted at 0.25 degree. Is it an interpolation for the ERA Interim 1x1 degree resolution data? If so, please remark it. Again, if the resolution for the synoptic wind field is an important feature, it seems to me that another reanalysis dataset would be more interesting for this work.

We appreciate your suggestions. We wanted to confirm the occurrence of the lake breeze at different levels and different time before. But it makes more sense to compare the variables at the same level and different time as you commented. We have replotted figure 3 shown below. In order to show clearly, the geopotential height lines and its labels are presented in black.

[Figure]

Fig. 3. Synoptic background on July 29, 2012. Blue boxes represent the two lakes area. (a) and (b) show the wind field (vector arrow), temperature field (color-filled contour), and geopotential height field (black lines) at 500 hPa (~5500 m) at 12:30 LT (Fig. 3a) and at 18:30 LT (Fig. 3b). The vertical wind (*w*, Figs. 3c and 3d) and the zonal wind (*u*, Figs. 3e and 3f) below 500 hPa are also shown

It seems that the comment (1) and comment (6) (iii) are related. As the answer for comment (1), we used the ERA-Interim Reanalysis Data with a 0.25° × 0.25° resolution for the zonal and vertical winds in Fig. 3, and 1° × 1° resolution for temperatures and geopotential heights. We paid more attention to the synoptic wind field and the circulations induced by the surface heat flux anomaly. We have replotted Fig. 3 using ERA-Interim reanalysis data with a 0.25° × 0.25° resolution for all variables. We revised the statement as: "...using the ERA-Interim reanalysis data with

a 0.25° × 0.25° resolution for the two lakes area...The southerly wind controlled the entire region at 500 hPa at 12:30 LT (Fig. 3a) then it became divergent flow at 18:30 LT (Fig. 3b)."

The profiles of the buoyancy production/destruction and the shear production term, and wind velocity shown in Fig. 7 are the results at 15:30 LT. According to your comments, we have added the time as: "…… which is from the contributions of the resolved (RES) and subgrid (SGS) eddies at 15:30 LT (Figs. 7a, 7b, and 7c)"

(ii) I suggest using the designation "Buoyancy production/destruction" instead "Buoyancy flux" to refer to the turbulent kinetic energy (TKE) budget equation term to avoid any misinterpretation. Furthermore, "Buoyancy flux" is not precise to describe it on a physical sense.

Thanks for your suggestion. We have changed the "buoyancy flux" to "buoyancy production/destruction" for the title and the caption in figure 7.

(iii) The buoyancy production/destruction profiles showed a similar behavior for wind/no wind simulations in a homogeneous surface. However, for heterogeneous surface simulations, it is clear that resolved heat flux reaches a lower maximum and decreases differently from pure-convection (no wind) simulation. Could you briefly explain it on the text?

Thanks for your suggestion. We have added more comments about the RES buoyancy production/destruction reaching a lower maximum for wind simulations in the text as following: "The buoyancy production/destruction in the TKE budget

equation is $B = \dfrac{g}{\overline{\theta_v}} \overline{w'\theta_v'}$. The RES buoyancy production/destruction profiles show that the lower maximum occurs for the wind simulations over the heterogeneous surfaces. It is because the larger positive buoyancy production/destruction decreases outside the patches (Fig. S4 in the supplement) due to the significantly weakened updrafts of the patch-induced circulations by the background wind. Comparing with no wind simulations (Fig. S4b, S4c), the buoyancy production/destruction over the patch/patches decreases for wind simulations. It is probably caused by the relatively warm air in a thermal internal boundary layer (TIBL) formed over the patch/patches (Fig. S5b, S5c) due to the abrupt change in surface heat flux (Mahrt, 2000) with air flowing from the warm patch to the cold patch. Similar with the results of Zhou et al. (2018) and Liu et al. (2020), the cold center of the TIBL (Fig. S5e, S5f) moves to the downwind of the lake patches. "

*Liu, R., Sogachev, A., Yang, X., Liu, S., Xu, T., Zhang, J. 2020. Investigating microclimate effects in an oasis-desert interaction zone. Agricultural and Forest Meteorology, 290, 107992.*

*L. Mahrt, 2000. Surface Heterogeneity and Vertical Structure of the Boundary Layer. , 96(1-2), 33–62. doi:10.1023/a:1002482332477*

*Zhou, Y., Li. D., Liu, H. and Li, X.: Diurnal variations of the flux imbalance over homogeneous and heterogeneous landscapes. Boundary-Layer Meteorology, 168:417–442. https://doi.org/10.1007/s10546-018-0358-2, 2018.*

[Figure]

Fig. S4. The y-z cross sections of the buoyancy production/destruction (contour) with superimposed wind vectors composed of *v* and *w* wind over (a, d) homogeneous and (b, e, c, f) heterogeneous surfaces with (d, e, f) and without (a, b, c) background flow. Black lines on the x-axis represent the lake patches

[Figure]

Fig. S5. The y-z cross sections of the virtual potential temperature (contour) with superimposed wind vectors composed of *v* and *w* wind over (a, d) homogeneous and (b, e, c, f) heterogeneous

surfaces with (d, e, f) and without (a, b, c) background flow. Black lines on the x-axis represent the lake patches

(iv) I suggest a new figure, like figure 6, to describe the effect of buoyancy TKE budget term. It could be included on the main paper or on the supplementary documentation. It would clarify the interaction of lake patches with atmospheric flow and how it impacts the TKE balance.

Thanks for your suggestions. We have added a figure (Fig. S3) in the supplementary documentation, which shows the y-z section of the buoyancy production for runs with and without background wind over homogeneous and heterogeneous surfaces. We have added analyses about the interaction of lake patches with atmospheric flow as: "The buoyancy production/destruction in the TKE budget equation is $B = \frac{g}{\overline{\theta_v}} \overline{w'\theta_v'}$. The RES buoyancy production/destruction profiles show that the lower maximum occurs for the wind simulations over the heterogeneous surfaces. It is because the larger positive buoyancy production/destruction decreases, especially in the downstream of the patch (Fig. S4 in the supplement), which is due to the significantly weakened updrafts of the patch-induced circulations by the background wind. Comparing with no wind simulations (Fig. S4b, S4c), the buoyancy production/destruction over the patch/patches decreases for wind simulations. It is probably caused by the relatively warm air in a thermal internal boundary layer (TIBL) formed over the patch/patches (Fig. S5b, S5c) due to the abrupt change in surface heat flux (Mahrt, 2000) with air flowing from the warm patch to the cold patch. Similar with the results of Zhou et al. (2018) and Liu et al. (2020), the cold center of the TIBL (fig. S5e, S5f) moves to the downwind of the lake patches."

(v) Is the wind shear the source of SGS shear production peak at $z/z_i = 0.6$ or is there an unusual feature on the momentum flux profile? Again, a plot with the momentum

flux profile (resolved and SGS) could be presented on the supplementary
documentation to clarify it.

Thanks for your suggestion. Corresponding to the Fig.7b, we have plotted a figure
of the momentum flux profiles as Fig. S3 added in the supplement. It confirms that
the wind shear causes the peak of the SGS shear production at $z/z_i = 0.6$. Because the
larger momentum above $0.6 \, z_i$ is transported downward, which leads to the increase of
the wind speed and wind shear below. We have added comments as: "In addition,
wind shear is the source of the SGS shear production peak at $z/z_i = 0.6$ due to the
larger momentum flux above $0.6 \, z_i$ (see Fig. S3) increasing the wind speed and wind
shear below."

[Figure]

Fig. S3. Vertical profiles of the momentum flux for runs HOMW, A1LW, and A2LW with
background flows. The resolved and subgrid results are presented as red and blue lines,
respectively

(vi) The wind profiles presented an interesting feature on heterogeneous surface
simulations. In the homogeneous case, the wind profile seems to be log-linear close to
surface and showed a clear mixed layer above it. For the lake simulations, the wind
profiles exhibit a feature similar to a stable boundary layer, with a maximum local
wind. It is an interesting feature that could be better discussed on the text. I suggest
plotting the potential temperature profile associated to these wind profiles to better
understand the PBL vertical structure at this time. One could say that an internal
boundary layer process would be occurring here. Furthermore, I suggest plotting a

log-linear law and the geostrofic wind components with wind profiles to better visualize and discuss these wind profiles.

We appreciate your detailed comments. We have replotted the Fig. 7d with including log-linear wind profiles and virtual potential temperature profiles associated with these wind profiles. In order to illustrate the effects of the geostrophic wind on the TIBL, wind profiles from runs with no geostrophic wind (runs A1LNG and A2LNG) have been added in the Fig. 7d. A constant geostrophic wind profile is used in this study. Considering the geostrophic wind components have been shown in the Fig. 2g, they are not plotted in the Fig. 7d. The following text is added: "Fig. 7d showed the wind profiles (red lines) for runs with background wind (HOMW, A1LW, A2LW) and without geostrophic wind (A1LNG, A2LNG), and the virtual potential temperature profiles (blue lines). It shows that patch-induced circulations reduce the modeled mean wind speed below the height of about 800 m, for the largest wind speed exists in the homogeneous case (red solid line). The wind profile is log-linear below the height of 20 m and shows a clear mixed layer above it for the homogeneous run, which correspond to a mixed layer shown by the virtual potential temperature profile (blue solid line). For the one/two-lake simulations, the wind profiles (red dotted and dashed lines) exhibit a feature of a stable boundary layer (blue dotted and dashed lines) with a maximum local wind at about 400 m. It should be noted that the stable stratification of wind profiles between 200 m and 1000 m are probably caused by the process of the TIBL. It is confirmed by the similar wind profile features from the runs without geostrophic wind over the heterogeneous surface."

[Figure]

Fig. 7. Vertical profiles of (a) the buoyancy production and (b) the shear production term for runs HOMW, A1LW, and A2LW with background flows, and (c) the profiles of the buoyancy flux for runs HOM, A1L, and A2L without background flows. (d) The simulated horizontal wind (red lines) versus logarithm of height for runs HOMW, A1LW, A2LW, A1LNG and A2LNG, and the virtual potential temperature profiles (blue line) at this time. The resolved and subgrid results are presented as red and blue lines in (a), (b) and (c), respectively. The black lines in (b) are the total (resolved and subgrid scale) shear production term

(vii) Regarding the possible internal boundary layer formation, an extra plot for potential temperature, similar to figure 6, could be made.

Thanks for your suggestion. The vertical distribution of the potential temperature is plotted as Fig. S5 in the supplement. Please find the detailed replay in the comment 7 (vi).

(8) Page 22, line 408 – I wonder if the weaker updrafts could explain the buoyancy TKE

budget term features highlighted previously. What do the authors think about that? If these two characteristics are related, please detail it on the text.

Thanks for your reminding. The weaker updrafts for wind simulations also correspond to the weaker buoyancy production term in the analysis of Fig.7. We also added the statement like: "...which is confirmed by the weaker buoyancy production/destruction for the heterogeneous simulations in the Figs. 7a and 7c."

(9) Page 23, figure 8 – I think it would be interesting to add an extra plot here with the homogeneous cases. It helps to evaluate the lake patch impact on the local circulation.

Thanks for your suggestion. We have added the instantaneous y-z cross sections of the vertical velocity and wind vectors above the homogeneous surfaces for runs without (Fig. 8g) and with (Fig. 8h) background wind. We also added the statement, like: "Comparing with the spatial distribution of the vertical velocity over the homogeneous surface (Figs. 8g and 8h), the lake patch/patches alters both the boundary-layer convection intensity and the local circulation."

[Figure]

Fig. 8. Instantaneous y-z cross sections of the vertical velocity (m s$^{-1}$) and wind vectors above the heterogeneous surfaces for runs (a and b) without and (c and d) with background wind, and (e and f) with the geostrophic wind removed, as well as the results over homogeneous surfaces for runs with (h) and without (g) background flow. The blue lines on the x-axis represent the lake patches

(10) Page 432, figure 10 – What time is it on the simulation? Is it on the same time of wind profiles from figure 7? I am asking it because the negative heat flux on the wind-simulations, above the lake patch, could be decreasing the turbulent viscosity and increasing the wind speed consequently. What do the authors think about that? It is important to note that the minimum flux value (negative) is in a magnitude so strong as it is close the surface. Furthermore, it happens around the same height of local maximum wind. I suggest plotting the potential temperature associated to these heat fluxes to better understand it.

Thanks for your comments. Yes, it is at the same time of wind profiles from Fig. 7 We have added the time in the caption of Fig. 10. The wind profiles in Fig. 7d are the domain-averaged. We have plotted mean profiles of the potential temperature and

wind speed above the lake patch shown in the following Fig. R3. It shows that the wind speed over the patch increases significantly for the wind simulations, and the height of the largest wind speed corresponds to height of the minimum heat flux over the lake patch/patches (at about 0.3 $z_i$ in Fig. 10d). The downward transport of the heat flux inhibits the turbulence mixing over the lake patch, and increases the wind speed consequently. This probably contributes to the local maximum wind speed. We have added the following text: "Notice that the minimum heat flux (at about 0.3 $z_i$) is in the magnitude so strong as it is close to the surface (0.1 $z_i$). It probably contributes the local maximum wind speed."

[Figure]

Fig. R3 The profile of the simulated potential temperature (left) and wind speed (right) over lake patch with (red and black lines) and without (blue and green lines) background wind at the same time of Fig. 10 and Fig. 7

(11) Page 26, figure 11 – This figure shows a clear transition between the land-lake PBL. I would like to see the wind speed of homogeneous simulation to compare it with the heterogeneous ones, as well as a comparison for potential temperature and heat flux. It could be interesting to understand a possible internal boundary layer formation.

Thanks for your suggestion. We have added the wind speed near the surface from the homogeneous simulations, and the potential temperature and sensible heat flux in Fig. 11. We have added the following statement about the TIBL, as: "Moreover, the

potential temperature (Figs. 11e, 11f) and the sensible heat flux (Figs. 11g, 11h) increase abruptly from the lake patch to the grass patch (e.g. from y=15 km to y=25 km in Fig. 11e), which indicates the formation of the TIBL."

[Figure]

Fig. 11. Variations in the (a and b)wind speed, (c and d) Reynolds stress, (e and f) potential temperature and (g and h) heat flux in the horizontal direction below 200 m for the cases with (blue and green lines) and without (yellow and red lines) background flows over homogeneous (red and green lines) and heterogeneous (blue and yellow lines) surfaces. Black lines on the x-axis represent the lakes

---

## Author Comment (AC4)

**Response to anonymous Referee #3**

We would like to thank the reviewer for taking the time to make detailed and useful comments. Details of the changes made are now given in the supplement in reply to your comments made one by one.

**Specific comments**

(1). The authors use the quasi-three-dimensional large eddy simulations (LES) in this study. The horizontal model domian of 135 km x 30 km with the mesh grid of 200 m. My question is why did not the author use the higher LES resolutions in this study? I wonder if the resolution of 200 m is appropriate for a LES study? In addition, what is the timestep in the simulations?

Thanks for your comments. A large horizontal domain (135 km  $\times$  30 km) is used to include possible mesoscale circulation due to the surface heat flux anomaly in this study. Considering the high computational cost, we employed a grid spacing of 200 m for the LES simulations. However, previous studies about the turbulence gray zone confirms the spatial resolution of 200 m in our study is appropriate.

Honnert et al. (2011) defined a dimensionless mesh size  $\Delta x/(z_i+z_c)$  to quantify the resolved and subgrid parts of the turbulence at different scales of any free convetice boundary layer, where  $z_i$  and  $z_c$  are the ABL height and the depth of the shallow cloud layer, respectively. Honnert et al. (2011) found that the resolved and subgrid TKE are equal for  $\Delta x/(z_i+z_c) = 0.2$ . In our study, the CBL height reaches to about 700 m at 09:30 and up to 1900 m at 18:30 (Fig. 4 in the main text). Clouds developed from a cloud base approximately 1000 m at about 12:30 (Fig. R1 is shown following).

Fig. R1 The heights of cloud base for the different runs.

Fig. R2 The partition of the LES, near gray-zone, gray-zone and mesoscale (Shown in Fig. 4 from Honnert et al. (2020))

The calculated dimensionless mesh size is about 0.12 at 12:30, and about 0.08 at 15:30, which indicates the resolved TKE is larger than the subgrid part, especially for the time of 15:30 (as Fig. R2). According to Honnert et al (2020), the CBL gray zone is roughly at 200 m  $\leq \Delta x \leq 2$  km when LES converging simulations is achieved at  $\Delta x \sim 20$  m with taking  $z_i = 1000$  m. It illustrates that the horizontal resolution of 200 m in our simulation lies in the near gray zone during the early CBL development (12:30), but is an appropriate resolution for the time of 15:30.

There are LES studies with the horizontal grid spacing equaling to or larger than 200 m investigating the CBL turbulence over the heterogeneous surface (Huang et al. 2010; Rai et al. 2016; Xu et al. 2018). For example, Huang et al. (2010) used the Met Office Large Eddy Model with grid spacings of 200 m to simulate the effects of

surface heat flux anomalies on the formation of deep boundary layer over the Sahara dessert.

We have clarified the resolution choice, as: "According to Honnert et al. (2011) and Honnert et al. (2020), the horizontal resolution of 200 m is reasonable in this LEM study."

The time step of 0.01s is applied for all simulations in this paper. According to your suggestion, a brief comment is added: "The time step is 0.01s for all simulations."

Honnert, R., Masson, V., & Couvreux, F. (2011). A diagnostic for evaluating the representation of turbulence in atmospheric models at the kilometric scale. J. Atmos. Sci., 68, 3112-3131.

Honnert R, Efstathiou G A, Beare R J, et al. The Atmospheric Boundary Layer and the "Gray Zone" of Turbulence: A Critical Review[J]. Journal of Geophysical Research: Atmospheres, 2020, 125.

Xu H, Wang M, Wang Y, et al. Performance of WRF Large Eddy Simulations in Modeling the Convective Boundary Layer over the Taklimakan Desert, China[J]. Journal of Meteorological Research, 2018.

Huang Q, Marsham J H, Parker D J, et al. Simulations of the effects of surface heat flux anomalies on stratification, convective growth, and vertical transport within the Saharan boundary layer[J]. Journal of Geophysical Research Atmospheres, 2010, 115.

Rai R K, Berg L K, Kosovi B, et al. Comparison of Measured and Numerically Simulated Turbulence Statistics in a Convective Boundary Layer Over Complex Terrain[J]. Boundary-Layer Meteorology, 2016, 163(1):1-21.

(2). The thermal internal boundary layer (TIBL) would form when cold air passes over the warm surface. It has been reported that a large scale convective TIBL could form due to the surface heterogeneity. If there exists a TIBL when the air flows from the cold lake patch to the warm grass land in your study? How does the TIBL affect the turbulence interaction over the heterogeneous surfaces in your simulations?

Thanks for your comment. The TIBL forms when the air blowing from the cold lake to the warm grassland in our study. According to the other reviewer's comment, we have plotted the vertical TKE distribution for the runs with wind in Fig. 6d, 6e, 6f. We have added the statement about how the TIBL affect the turbulence interaction over the heterogeneous surfaces: "The similar TKE distribution occurs when the background wind exists over the homogeneous surface (Figs. 6a and 6d). It should be noted that there is larger TKE over the patch/patches (below 0.2  $z_i$ ) as the similar pattern of TKE in Papangelis et al. (2021), from which the TIBL can be recognized (Figs. 6e and 6f). The background wind tends to reduce the TKE outside patch/patches while enhance it over the patch/patches. Moreover, the background wind inhibits the development of the patch-induced circulation because the divergent wind derived from the heterogeneous surface can not be viewed at 15:30."

Papangelis G, Tombrou M, J Kalogiros. The Saharan convective boundary layer structure over large scale surface heterogeneity: A large eddy simulation study[J]. Atmospheric Research, 2020, 248:105250.